🔓 | **Open Peer Review** | Environmental Microbiology | Research Article

# Evaluating the legacy of drought exposure on root and rhizosphere bacterial microbiomes over two plant generations

A. Fina Bintarti,[1] Abby Sulesky-Grieb,[2] Joanna Colovas,[2] Brice Marolleau,[3] Tristan Boureau,[3] Marie Simonin,[3] Matthieu Barret,[3] Ashley Shade[1]

**ABSTRACT** Drought is a critical risk for staple crops like common bean (*Phaseolus vulgaris* L.). We conducted an experiment to understand the legacy effects of repeated drought exposure across plant generations on the root and rhizosphere microbiome of the common bean, hypothesizing that a legacy of exposure improves overall plant microbiome resilience. We profiled the bacterial microbiome using marker gene amplicon sequencing over two plant generations in a complete factorial design for two common bean genotypes, Red Hawk and Flavert. We performed parallel experiments for Red Hawk in two different countries using soils of Pays de la Loire, France, and Michigan, USA. Despite the clear and relatively consistent drought effects on the plant phenotypes, there was neither a strong response of the Red Hawk microbiomes to drought nor a notable legacy of drought exposure. For Flavert, there was a minor legacy drought effect for the second generation in the rhizosphere microbiome beta diversity, while its root had no legacy effect observed. This study demonstrates that below-ground plant microbiomes can be resistant to drought stress and that cross-generational legacy depends on soil origin and host genotype. Such parallel experiments across countries are useful to inform generalities and build theory toward prediction on microbiome responses to global change.

**IMPORTANCE** Drought remains an important challenge in crop agriculture because of climate change, and plant microbiome management has potential to support plant resilience to drought. Here, we investigated the impact of drought and drought legacy across two generations on the root and rhizosphere microbiomes of the drought-susceptible legume common bean, a key staple food crop with production widely distributed across the Americas, Africa, Europe, and Asia, and which is of critical importance for food security in many of its production regions. Despite host plant decline with drought, the effects of drought on the microbiomes were either not observed, inconsistent, or weak, suggesting overall microbiome resistance and limited drought legacy. This work provides insights into how the stability of the below-ground plant microbiome can be driven by stress resistance, offering a different perspective on how the microbiome could be managed to support crops facing drought.

**KEYWORDS** plant microbiome, ecological resistance, water limitation, amplicon sequencing, disturbance ecology, rhizobiome

Agricultural systems must now regularly contend with severe and sometimes unpredictable environmental stress (1–3). Changing weather patterns have resulted in droughts in many regions, which are particularly devastating in countries that face food insecurity (4–8). An important and growing area of research is focused on the impact of these environmental changes on the microbiomes of crops (9–12), which comprise millions of bacterial, archaeal, and fungal cells that inhabit plant surfaces and

**Peer Reviewer** Aqleem Abbas, Huazhong Agriculture University, Wuhan, China

Address correspondence to Ashley Shade, ashley.shade@cnrs.fr.

A. Fina Bintarti and Abby Sulesky-Grieb contributed equally to this article. Co-first authorship determine alphabetically.

The authors declare no conflict of interest.

See the funding table on p. 17.

internal tissues, and colonize the soil proximal to roots (13, 14). The beneficial members of the microbiome play an essential role in plant health in a changing climate (15–17). They collectively also provide vital agricultural services; for example, they support water and nutrient assimilation for plants, the biogeochemical cycling of nutrients in soil and root system, and plant pathogen resistance (13, 14, 18–22). While many studies have investigated the impacts of environmental stress on the plant microbiome (23–29), very few have studied the effects of repeated stress and over multiple generations (30, 31). Repeated stress exposure that occurs over multiple growing seasons and plant generations is expected to have negative impacts on crop agriculture, for example, by reducing seed quality and degrading soil quality (1, 2). Compounded drought stress is of particular concern with climate change, as repeated exposure to drought is expected to become more frequent for many production areas.

Our study aimed to understand the implications of repeated drought on the microbiome of the legume common bean (*Phaseolus vulgaris* L.). Specifically, we were interested in the legacy of multi-generational drought exposure on the below-ground microbiome and defined legacy as enduring or residual effects of the stress on the compositional selection or recruitment of microbial taxa by the host environment. Legumes provide valuable nutrition and perform symbiotic nitrogen fixation, providing usable nitrogen to other crops (32). Legume services are particularly valuable in developing economies that are most significantly impacted by environmental stress due to a lack of infrastructure to manage poor growing conditions and reliance on subsistence farming. Common beans are a vital food source worldwide, with 27.5 million metric tons of food produced annually and cultivated across 34.8 million hectares of land as of 2020 (33, 34). *Phaseolus vulgaris* L. includes a wide range of "dry edible bean" and fresh "garden bean" crops, including kidney beans, black beans, navy beans, green beans, and others (35). These varieties have been selected over 8,000 years of domestication from two ancestral lines in Mesoamerica and the Andes Mountains (33, 36). Furthermore, the ability to continue to produce dry beans in developing economies is vital for food security (37).

Breeding has been done to improve the drought tolerance of common bean varieties, but managing the plant microbiome may also offer some options to improve plant performance during drought (38, 39), including how to decrease any potential legacy effect of multi-generation exposure to drought. Various studies have been conducted to understand the bean rhizosphere (40, 41) and seed (42–44) microbiome, their association with nitrogen-fixing rhizobia (45–47), and the impact of abiotic stress, such as drought, on the bean microbiome (31, 48, 49). Our previous research on the seed endophytic microbiome suggested that there was limited drought legacy effect on this particular compartment (31), while our other work showed that particular rhizosphere microbiome members were activated in response to drought only in the plant environment, and not in droughted soil in the absence of the plant (49). However, it remains unclear how repeated drought across generations may affect the below-ground microbiome of common bean, and this effect may differ from that on the seed due to the potentially stronger influence of environmental recruitment on below-ground microbiome assembly.

To address this knowledge gap, we conducted a two-generation experiment in which we exposed common bean plants to drought in agricultural soil under controlled growth chamber conditions. We first aimed to confirm that (i), under our experimental conditions, the drought negatively impacts the bean plant health. We then hypothesized that (ii), within a generation, the drought reduces the bacterial root and rhizosphere microbiome alpha diversity and increases its variance (as a microbiome disturbance expectation aligned to the Anna Karenina hypothesis, e.g., references 50, 51), and (iii) that the repeated exposure to drought over two generations has compounded legacy consequences for the plant and its microbiome in structure (beta diversity) and variance (dispersion). To assess the impact of drought on the species more broadly, we tested our hypotheses in two different genotypes of *Phaseolus vulgaris* L.: the Red Hawk variety,

a kidney bean cultivar bred in North America, and Flavert, a European Flageolet bean cultivar (52–54). We included production soils from two distant geographic locations where beans are grown—Pays de la Loire, France, and Michigan, USA—to determine how the location shapes drought response.

## RESULTS

### Overview

The experimental design is shown in Fig. 1. The experiment started with seeds from a Generation 0 (G0) seed pool, which we germinated and then exposed to either a well-watered condition or a drought condition in which water was decreased by 66% (Fig. S1.1). We harvested seeds from Generation 1 (G1) plants and used them to grow Generation 2 (G2), in which the plants either received the same or opposite treatment as in G1, with the parent lines tracked and distributed across the G2 experimental conditions. The plant growth, yield, and 16S V4 rRNA gene amplicon sequencing of the root and rhizosphere microbiome were analyzed to test our hypotheses.

Because the two locations had several necessarily differing parameters (see "Materials and Methods" section), differences in their microbiomes were expected. The datasets from Michigan and Pays de la Loire were quality controlled and analyzed independently, and the comparative analyses were focused on overall plant and microbiome trends and dynamics. Our objective was to assess ecological patterns and generalities in consecutive, multi-generational drought response in the microbiome despite differences due to different soils and geographic locations.

### Plants responded differently to drought in different locations

Plant biomass, yield, and photosynthetic data were collected. For plants grown in Pays de la Loire, France, with a slightly longer stress period, both Red Hawk and Flavert showed a decrease in the number of pods on the drought-treated plants (Welch's two-sample $t$-test, Red Hawk: $t = 4.26$, $P = 0.004$; Flavert: $t = 7.06$, $P = 0.0001$) (Fig. 2, A.1; Tables S2 and S3). However, the decrease in pod number did not correspond to a decrease in seed number (Fig. 2, A.2). For Red Hawk beans grown in Michigan, the numbers of pods, seeds, and root biomass were not affected by drought (Fig. 2, B.3, B.4, B.6; Table S2). However, the photosynthetic rate (Welch's two-sample $t$-test: $t = 3.28$, $P = 0.01$), stomatal conductance (Welch's two-sample $t$-test: $t = 2.82$, $P = 0.02$), and above-ground mass (Welch's two-sample $t$-test: $t = 2.49$, $P = 0.03$) were reduced in the drought plants compared to the control plants (Fig. 2, B.1, B.2, B.5; Table S2). These results generally indicate that there was a negative effect of drought on the G1 plants, but that the precise effect was inconsistent across the different locations.

There was a strong effect of G2 drought on the Flavert (two-way ANOVA, pod count: $F = 85.63$, $P = 7.99e-08$; seed count: $F = 85.79$, $P = 7.89e-08$, shoot mass: $F = 161.86$, $P = 8.78e-10$) and Red Hawk (two-way ANOVA, pod count: $F = 28.13$, $P = 7.14e-05$; seed count: $F = 141.75$, $P = 2.31e-09$, shoot mass: $F = 225.98$, $P = 7.4e-11$; root mass: $F = 21.95$, $P = 0.0002$) plants grown in Pays de la Loire (Tables S4 and S5). However, the observed G2 drought effect was independent of the G1 condition. Specifically, the number of pods, seeds, and above-ground biomass was decreased for both bean genotypes grown in Pays de la Loire (Fig. 3A). In contrast, the root mass of Red Hawk in Pays de la Loire was increased by drought (Fig. 3, A.4). For Red Hawk plants grown in Michigan, only the photosynthetic rate (two-way ANOVA: $F = 18.43$, $P = 0.0006$) and stomatal conductance (two-way ANOVA: $F = 46.8$, $P = 5.59e-06$) were decreased by drought, similar to what was observed during the G1 drought (Fig. 3, B.1, B.2; Table S4). These data indicate that the Flavert and Red Hawk plants in Pays de la Loire were more negatively impacted by the drought in G2 than the plants in Michigan. The legacy of G1 condition did not affect most plant outcomes in G2, except for Red Hawk biomass in Pays de la Loire. Flavert plants that were droughted in both generations had lower above-ground biomass than plants that were not droughted in G1 (Fig. 3, A.3).

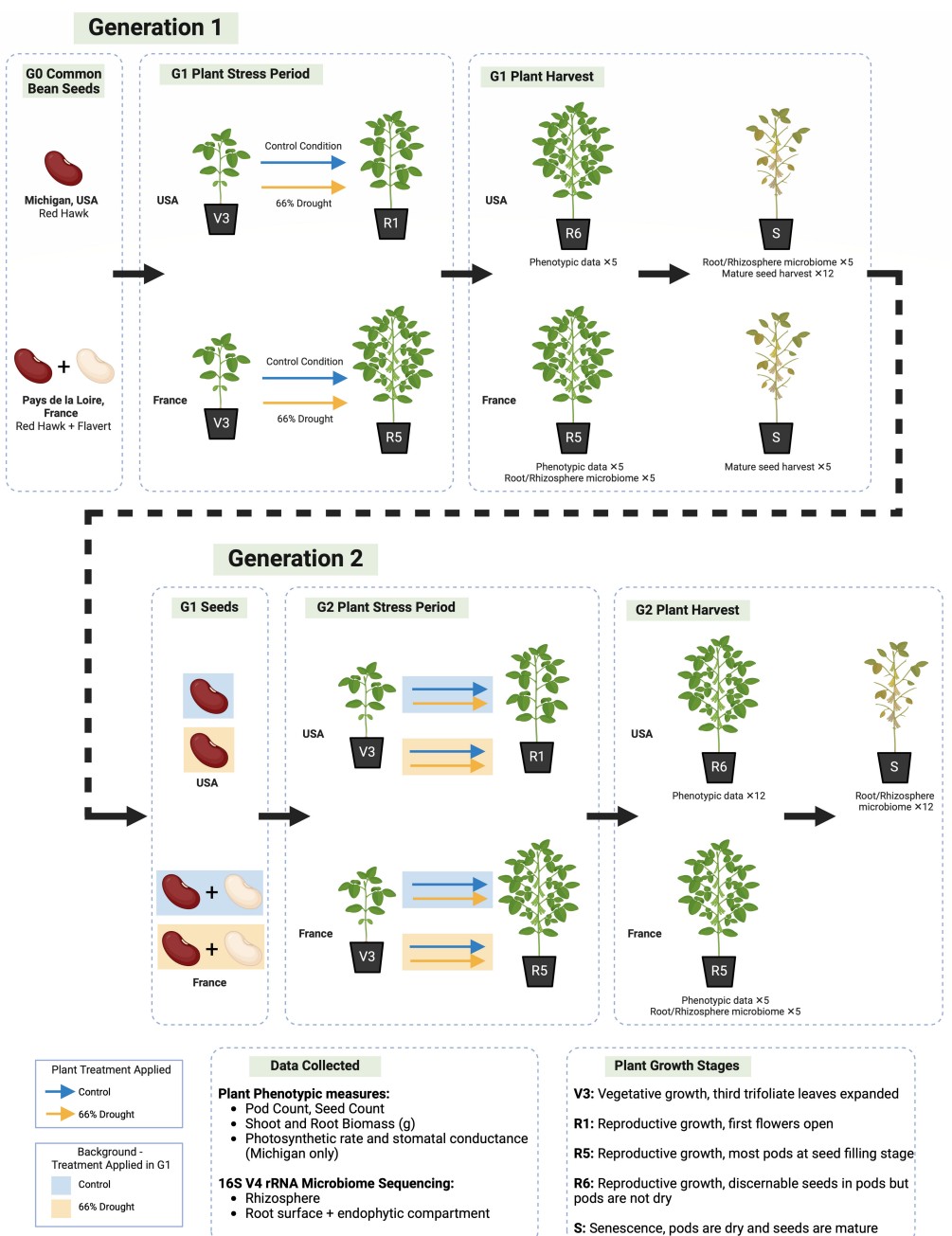

**FIG 1** Experimental design. *Phaseolus vulgaris L.* (common bean) plants were grown over two generations in a factorial design under either control treatment with ample water and nutrients, or drought treatment with 66% less water and equal nutrient concentration to control plants. The common bean variety "Red Hawk" was grown in Michigan, and both "Red Hawk" and "Flavert" were grown in Pays de la Loire. Drought treatments were started at the V3 growth stage when plants had three trifoliate leaves expanded and were concluded at the R1 stage (first open flowers) in Michigan and at the R5 stage (half of the pods filled with discernible seeds) in Pays de la Loire. The replication indicated is per treatment, including the total number of plants used for microbiome analysis and for plant phenotyping. Image created in BioRender (Sulesky, A., 2025) https://BioRender.com/l13n679.

## Bacterial community alpha diversity is decreased by drought in Flavert plants

Overall, the bacterial community richness (number of observed ASVs) in Pays de la Loire microbiomes was lower than the alpha diversity observed in Michigan samples. Sequencing coverage was sufficient for both data sets, as the rarefaction curves reached

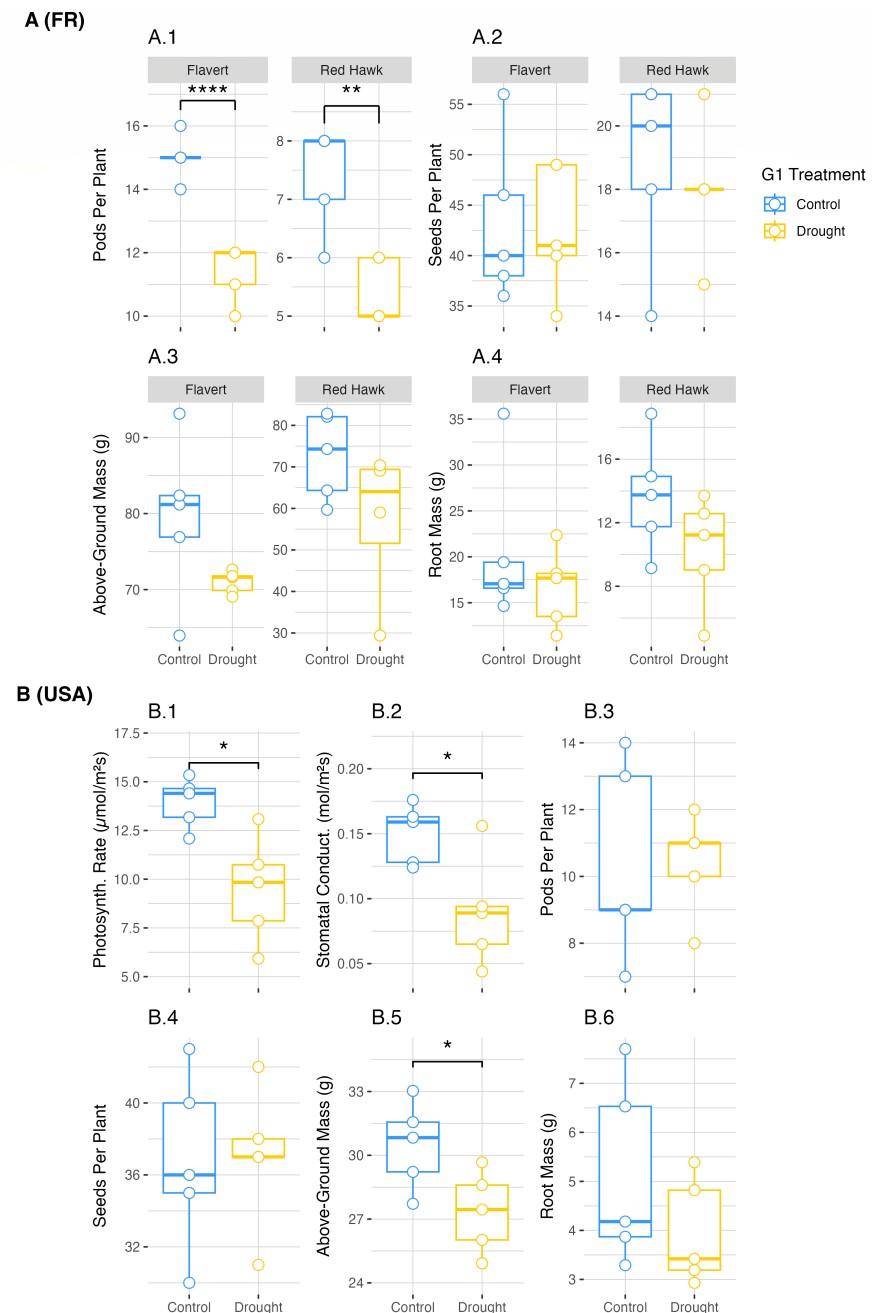

**FIG 2** Plant phenotypic measurements in G1. (A) Phenotypic measurements taken for common bean plants of both Flavert and Red Hawk genotypes grown in Pays de la Loire, France. Above-ground and root biomass measurements were taken on fresh plant tissue. (B) Phenotypic measurements were taken for common bean Red Hawk plants grown in Michigan, USA. Above-ground and root biomass measurements were taken on dry plant tissue. All above-ground biomass measurements include the total mass of stems, leaves, and pods. Photosynth. Rate = photosynthetic rate; Stomatal Conduct. = stomatal conductance. Two-way ANOVA with Tukey's HSD post-hoc test (Pays de la Loire, France) and Welch's two-sample $t$-test (Michigan, USA), * = $P$-value < 0.05, ** = $P$-value < 0.01, **** = $P$-value < 1e-4, $n$ = 5 plants per treatment.

asymptotes (Fig. S1). There were differences in richness between the two genotypes grown in Pays de la Loire in root (two-way ANOVA: $F$ = 7.651, $P$ = 0.0138; Tukey's HSD: P.adj < 0.05), but not in rhizosphere samples of drought-stressed plants in G1 (Fig. 4A and C; Table S6). However, these differences were not present in control plants. Global analysis revealed no effect of drought in G1 in both plant compartments in Pays de

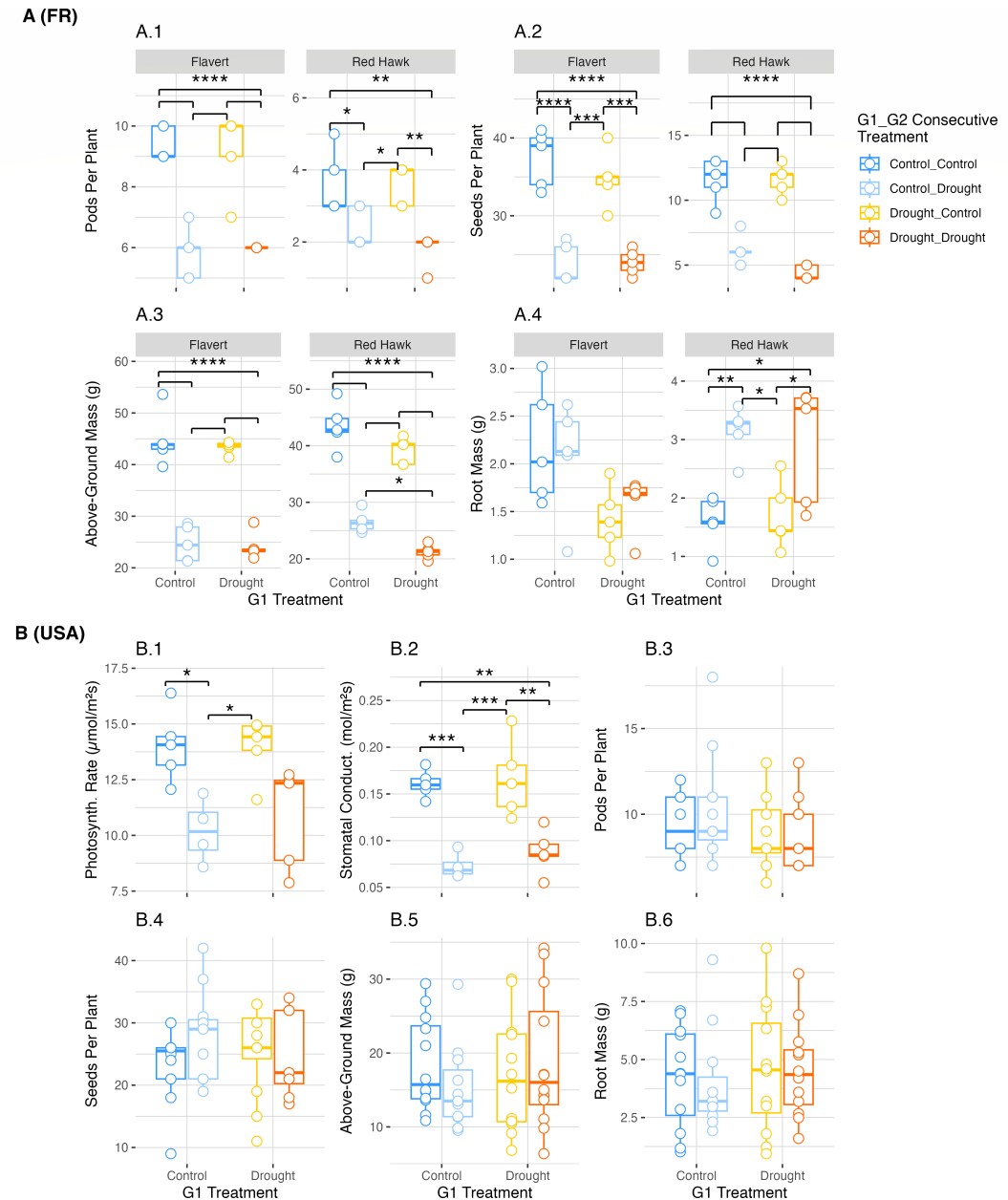

**FIG 3** Plant phenotype measurements in Generation 2. (A) Phenotypic measurements taken for common bean plants of both Flavert and Red Hawk genotypes grown in Pays de la Loire, France. Pays de la Loire had $n = 5$ plants per treatment. (B) Phenotypic measurements taken for common bean Red Hawk plants grown in Michigan, USA. Michigan had $n = 12$ plants per treatment. Above-ground and root biomass measurements were taken on dry plant tissue. Above-ground biomass measurements include the total mass of stems, leaves, and pods. Photosynth. Rate = photosynthetic rate; Stomatal Conduct. = stomatal conductance. Two-way ANOVA with post-hoc Tukey's HSD test, * = $P$-value < 0.05, ** = $P$-value < 0.01, *** = $P$-value < 0.001, **** = $P$-value < 1e-4. Non-annotated significance lines have the same p-value as lines above.

la Loire (Table S6). Similarly, the G1 microbiome alpha diversity of Red Hawk in Michigan was unaffected by drought (Welch's two-sample $t$-test, root: $t = 0.055$, $P = 0.958$; rhizosphere: $t = 0.792$, $P = 0.472$) (Fig. 4B and D; Table S6).

In the G2 experiment, we observed an effect of G2 drought on root microbiome alpha diversity in Pays de la Loire (three-way ANOVA: $F = 7.023$, $P = 0.0135$), but there was no legacy of G1 drought (three-way ANOVA, genotype × G1 × G2: $F = 0.03$, $P = 0.86$) (Table S7). Further post-hoc analysis showed no differences between groups for Flavert root microbiomes in Pays de la Loire, indicating the drought effect was marginal (Fig.

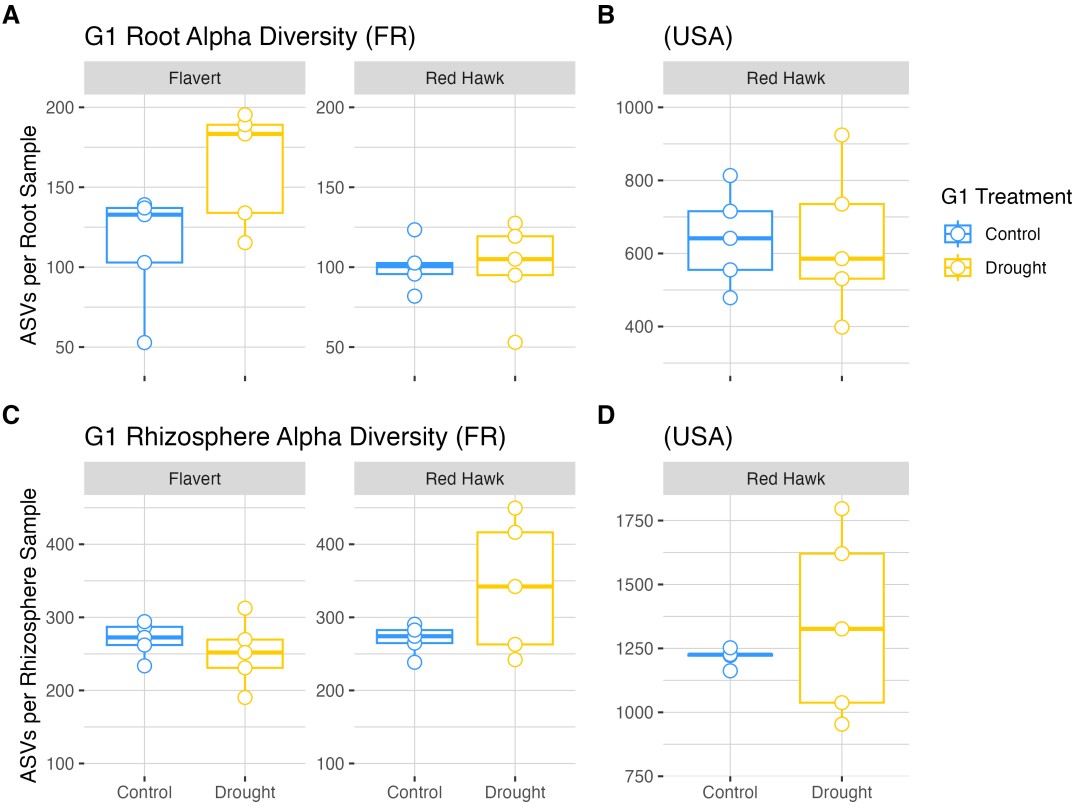

**FIG 4** Alpha diversity observed for root and rhizosphere samples in Generation 1. (A) Root microbiomes of plants grown in Pays de la Loire, France. (B) Root samples of plants grown in Michigan, USA. (C) Rhizosphere samples of plants grown in France. (D) Rhizosphere samples of plants grown in the USA. Plant genotype is indicated by bars above the panels. Two-way ANOVA with Tukey's HSD post-hoc test (Pays de la Loire, France) and Welch's two-sample *t*-test (Michigan, USA) ($n = 5$ per treatment). Absence of annotation indicates no significant difference (*P*-value > 0.05).

5A). There also was no legacy effect of drought detected for Michigan Red Hawk root microbiomes (two-way ANOVA, G1 × G2: $F = 1.45$, $P = 0.23$) (Fig. 5B; Table S7). The two genotypes grown in Pays de la Loire were different in their rhizosphere microbiome alpha diversity (three-way ANOVA: $F = 4.983$, $P = 0.03378$; Tukey's HSD: P.adj < 0.001) (Fig. 5C; Table S7). We observed a significant effect of G1 drought (three-way ANOVA: $F = 9.132$, $P = 0.00532$) on the richness of rhizosphere microbiome in Pays de la Loire (Table S7). Interestingly, the effects of drought were more pronounced in Flavert G2 plants that had experienced drought in G1, showing a greater decrease in richness compared to G1 control plants (Tukey's HSD, Control_Control vs Drought_Control: P.adj < 0.01; Control_Control vs Drought_Drought: P.adj < 0.01) (Fig. 5C; Table S8).

Overall, genotype was found to play a significant role in determining root and rhizosphere alpha diversity, and Flavert plants had decreased alpha diversity under drought conditions in G2, with no legacy effect detected. A global interaction between G1 and G2 treatments was initially detected in the rhizosphere microbiome of Red Hawk in Michigan (two-way ANOVA, G1 × G2: $F = 4.145$, $P = 0.047$) (Table S7), but it was not supported by post-hoc pairwise tests. Thus, a legacy of the G1 drought on the G2 outcome was not reflected in Red Hawk rhizosphere microbiome in either location, and, furthermore, the Red Hawk rhizosphere alpha diversity was not affected by the drought in G2 (Tukey's HSD: P.adj > 0.05) (Fig. 5C and D). Thus, despite the fact that Red Hawk plant traits were impacted by drought in Pays de la Loire, these plants' alpha diversity was unaffected.

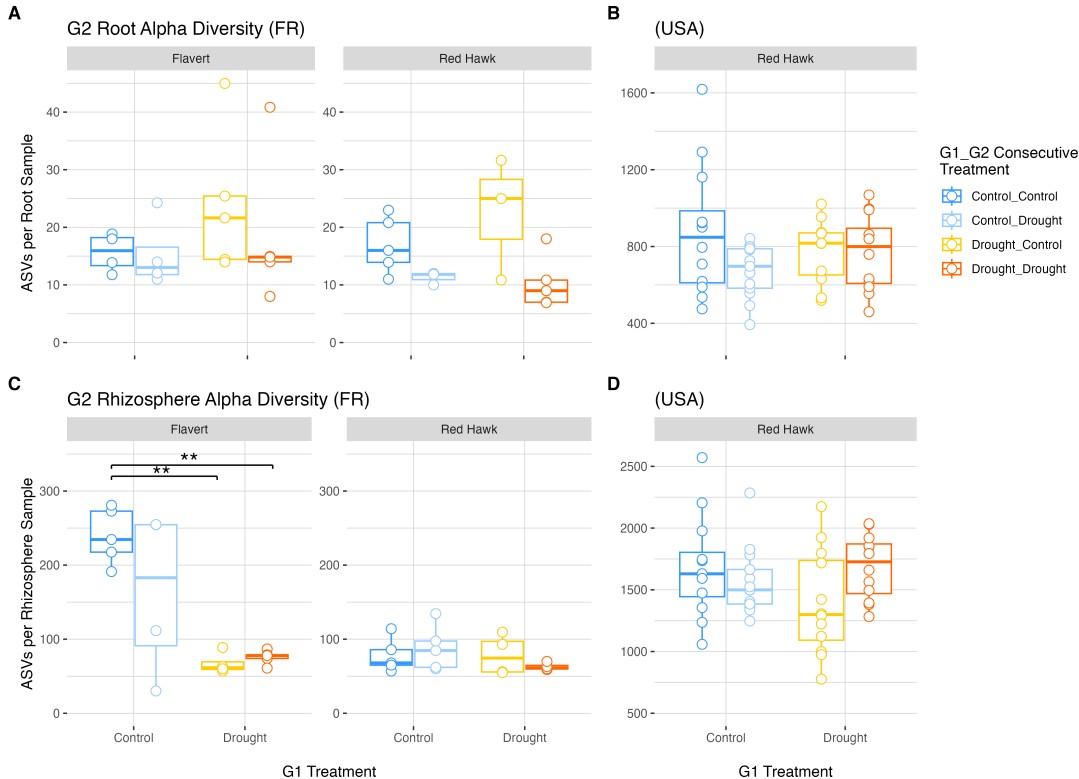

**FIG 5** Alpha diversity observed for root and rhizosphere samples in Generation 2. (A) Root samples of plants grown in Pays de la Loire, France (*n*=5 plants per treatment). (B) Root samples of plants grown in Michigan, USA (*n* = 12 plants per treatment). (C) Rhizosphere samples of plants grown in France. (D) Rhizosphere samples of plants grown in the USA. Plant genotype is indicated by bars above the panels. Two-way ANOVA with Tukey's Honest Significant Difference test, ** = *P*-value < 0.01.

## Inconsistent drought and drought legacy effects on bacterial community structure

Beta diversity of the root and rhizosphere microbial communities was analyzed independently for Pays de la Loire and Michigan plants. In G1, there were differences in the bacterial community structure of the two genotypes grown in Pays de la Loire in root (PERMANOVA: $R^2 = 0.49$, $F = 17.09$, $P = 0.0001$) (Fig. 6A and B; Table S9), but not in rhizosphere (PERMANOVA: $R^2 = 0.06$, $F = 1.22$, $P = 0.22$) (Fig. 6D and E) samples. When the genotypes were analyzed separately, the drought treatment in G1 had no influence on the root beta diversity in Pays de la Loire (PERMANOVA, Red Hawk: $R^2 = 0.11$, $F = 1.01$, $P = 0.39$; Flavert: $R^2 = 0.12$, $F = 1.05$, $P = 0.35$) (Table S10). Drought mildly affected the beta diversity in G1 rhizosphere (PERMANOVA: $R^2 = 0.09$, $F = 1.72$, $P = 0.02$), but not in root (PERMANOVA: $R^2 = 0.03$, $F = 1.07$, $P = 0.31$) (Table S9). We also found that the effect of drought on the rhizosphere communities in G1 differed by genotype. For example, when we analyzed separately, there were significant differences between drought and control in the rhizosphere beta diversity for Flavert (PERMANOVA: $R^2 = 0.16$, $F = 1.58$, $P = 0.02$) (Fig. 6D; Table S10), but not for Red Hawk (PERMANOVA: $R^2 = 0.13$, $F = 1.22$, $P = 0.1$) (Fig. 6E; Table S10) genotype. However, the drought treatment did not affect the root or rhizosphere beta diversity in Michigan plants in G1 (PERMANOVA, root: $R^2 = 0.15$, $F = 1.43$, $P = 0.14$; rhizosphere: $R^2 = 0.1$, $F = 0.97$, $P = 0.43$) (Fig. 6C and F; Table S9).

In G2, there were also differences between the two plant genotypes in Pays de la Loire for both compartments (PERMANOVA, root: $R^2 = 0.31$, $F = 15.47$, $P = 0.0006$ [Fig. 7A and B; Table S11]; rhizosphere: $R^2 = 0.06$, $F = 2.57$, $P = 0.0007$ [Fig. 7D and E; Table S11]). Flavert root beta diversity was affected by the interaction of the G1 × G2 treatments (PERMANOVA: $R^2 = 0.25$, $F = 4.94$, $P = 0.02$) (Fig. 7A; Table S12), indicating

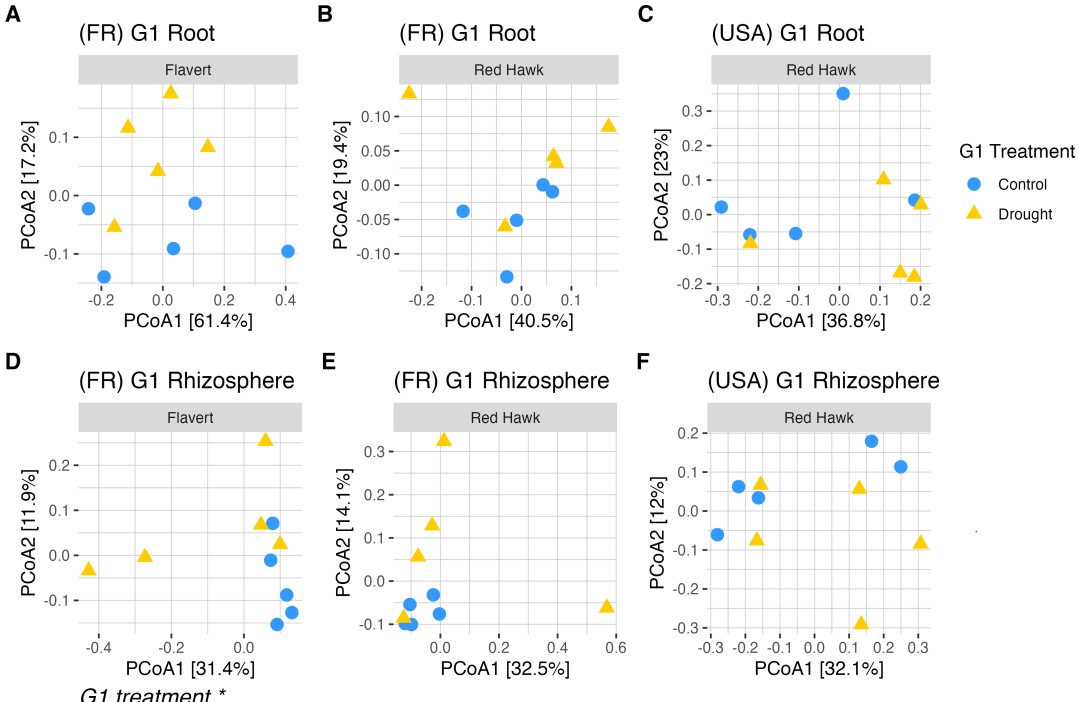

**FIG 6** PCoA ordinations of beta diversity Bray-Curtis dissimilarities in the Generation 1 (G1) experiment. (A and B) Root samples of Flavert and Red Hawk plants grown in Pays de la Loire, France. (C) Root samples of plants grown in Michigan, USA. (D and E) Rhizosphere samples of Flavert and Red Hawk plants grown in Pays de la Loire, France. (F) Rhizosphere samples of plants grown in Michigan, USA. There were significant differences between genotypes in root samples in France (PERMANOVA: $R^2 = 0.49$, $F = 17.09$, $P = 0.0001$). Plant genotype is indicated by the panel labels. * = $P$-value < 0.05, $n = 5$ per treatment.

that the effect of drought in the G2 was affected by its G1 treatment. Pairwise multi-level comparisons of Flavert's root microbiome showed distinct community structures between the Control_Drought and Drought_Drought plants (PERMANOVA: $R^2 = 0.42$, $F = 5.1$, $P = 0.018$). Red Hawk roots in G2 in Pays de la Loire were unaffected by G1 drought (PERMANOVA: $R^2 = 0.03$, $F = 0.41$, $P = 0.59$) and G2 drought (PERMANOVA: $R^2 = 0.06$, $F = 0.86$, $P = 0.38$) (Fig. 7B; Table S12). However, a mild effect of G2 drought was observed on Red Hawk roots in Michigan (PERMANOVA: $R^2 = 0.03$, $F = 1.25$, $P = 0.029$) (Fig. 7C; Table S11).

In the G2 rhizosphere samples from Pays de la Loire, the impact of drought was influenced by genotype, as demonstrated by a significant interaction between genotype and G1 treatment (PERMANOVA: $R^2 = 0.07$, $F = 2.75$, $P = 0.0006$) (Table S11). Flavert plants exhibited distinct rhizosphere communities based on G1 legacy condition (PERMANOVA: $R^2 = 0.2$, $F = 4.32$, $P = 0.0001$) (Fig. 7D; Table S12), while Red Hawk plants were not affected by either G1 or G2 condition. In comparison, Red Hawk rhizosphere communities in Michigan were not affected by either generation (PERMANOVA, G1 × G2: $R^2 = 0.017$, $F = 0.81$, $P = 0.18$) or condition (PERMANOVA, G2: $R^2 = 0.015$, $F = 0.73$, $P = 0.39$) (Fig. 7F; Table S11).

Often, disturbed systems exhibit relative increases in their variability (50). Thus, we also analyzed the dispersion of the beta diversity (beta dispersion) within and between growth conditions and generations. In G1 plants, a minor effect of drought was only observed in rhizosphere communities of Flavert in Pays de la Loire with increased dispersion (PERMDISP: $F = 4$, $P = 0.046$) (Fig. 8A; Table S13). In G2, root communities of Flavert plants in Pays de la Loire were unaffected by drought (PERMDISP: $P > 0.05$) (Table S14). Meanwhile, we found that rhizosphere communities of Flavert G2 in Pays de la Loire had altered beta dispersion (PERMDISP, G2: $F = 4.05$, $P = 0.048$; G1 × G2: $F = 4.12$, $P = 0.02$) (Table S14), with Control_Drought plants having higher dispersion than Control_Control in the root samples (pairwise PERMDISP: $P < 0.05$) (Fig. 8C). Beta dispersion in Red

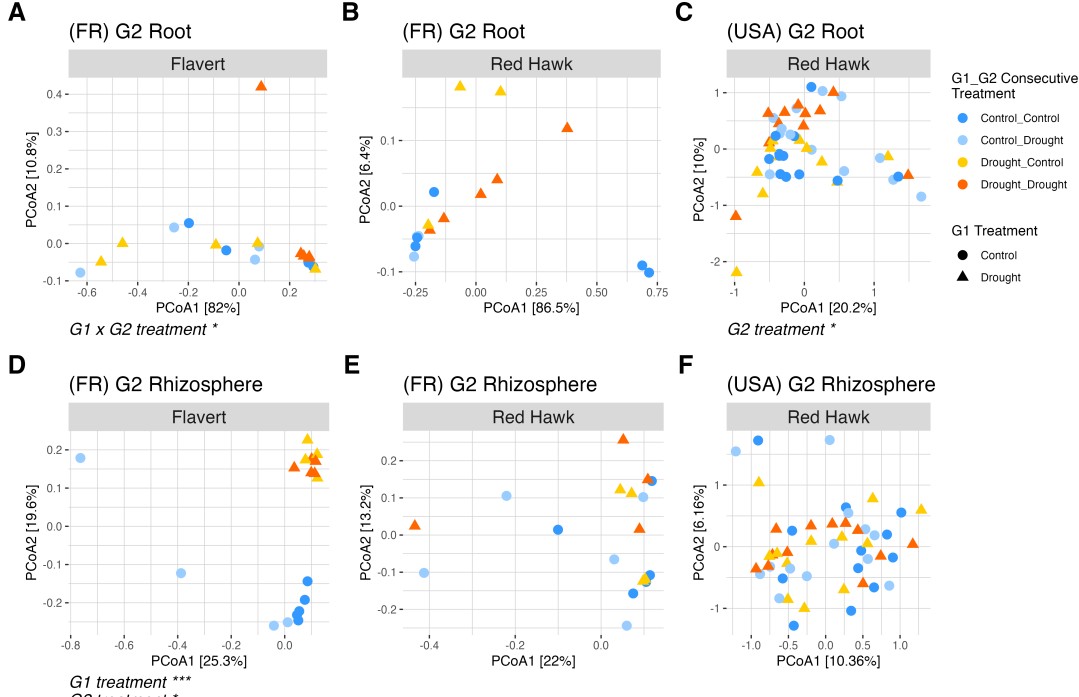

**FIG 7** PCoA ordinations of Bray-Curtis dissimilarities in the Generation 2 experiment (G2). (A and B) Root samples of Flavert and Red Hawk plants grown in Pays de la Loire, France. (C) Root samples of plants grown in Michigan, USA. (D and E) Rhizosphere samples of Flavert and Red Hawk plants grown in France. (F) Rhizosphere samples of plants grown in the USA. There were differences between genotypes in both root (PERMANOVA: $R^2$ = 0.31, $F$ = 15.47, $P$ = 0.0006) and rhizosphere (PERMANOVA: $R^2$ = 0.06, $F$ = 2.57, $P$ = 0.0007) samples in France. Plant genotype is indicated by panels labels * = $P$-value < 0.05, *** = $P$-value < 0.001. France $n$ = 5 plants per treatment; USA $n$ = 12 plants per treatment.

Hawk G2 plants was unaffected by drought in Pays de la Loire, despite there being a difference between Control_Drought and Drought_Control plants (pairwise PERMDISP: $P$ < 0.05, Fig. 8B). Red Hawk plants in Michigan had lower dispersion in the rhizosphere of droughted plants in G2, regardless of G1 treatment (pairwise PERMDISP: $P$ < 0.05). Consistent with the effects observed in alpha and beta diversity, the dispersion of the bacterial rhizosphere community in the Control_Drought Flavert plants was altered in G2, indicating the legacy effects for these plants. The Red Hawk rhizosphere beta dispersion was affected by the G2 drought in Michigan (Fig. 8D).

## DISCUSSION

Our study brings insights into the complex responses of a legume-associated, belowground microbiome to drought exposure, specifically testing for evidence of drought legacy across two generations. While drought negatively affected the plants in both locations, there was no strong evidence of a legacy effect on the microbiome, except mildly for the Flavert rhizosphere.

Across both generations and locations, Red Hawk's root and rhizosphere microbiomes generally were resistant (unchanged statistically) when faced with drought. Here, we apply the term "resistance" ecologically. Ecological stability includes two components: resistance and resilience (55–57). Resistance, the inverse of sensitivity, is the ability of a system to withstand change in the face of a disturbance. Resilience is the capacity of a system to recover after it has been changed by a disturbance. A system unchanged by a disturbance exhibits no statistically meaningful difference from the undisturbed condition. Furthermore, in the absence of change, resilience is not observable. Therefore, a highly resistant system is also a stable system.

Water limitation can directly shift microbiome structure and composition (58, 59) or indirectly affect it via plant-mediated physiological changes (60, 61). There are several

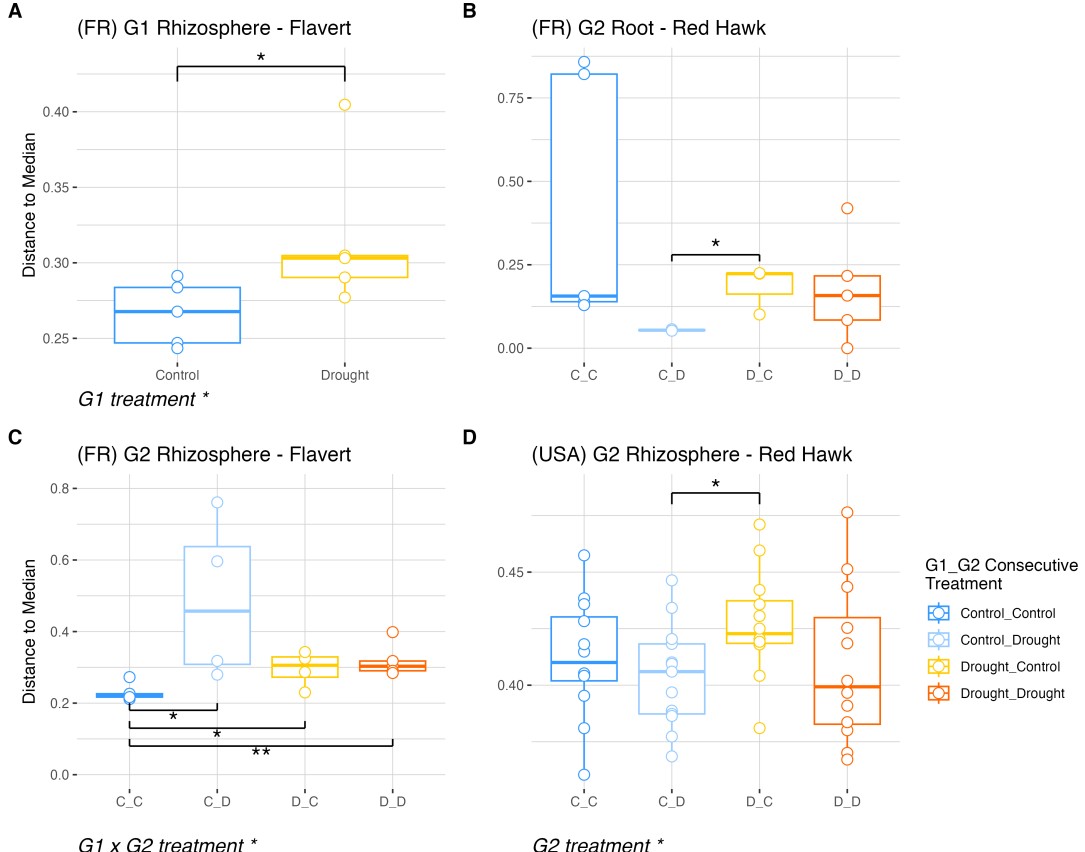

**FIG 8** Beta dispersion was calculated on the spatial median between samples. Only figures showing groups with statistically supported differences are shown. All other ordinations had no significant differences in dispersion between groups. (A) Distance to the median in G1 Flavert rhizosphere in Pays de la Loire, France. (B) Distance to the median in G2 Red Hawk roots in Pays de la Loire, France. (C) Distance to median in G2 Flavert rhizosphere in Pays de la Loire, France. (D) Distance to the median in G2 Red Hawk rhizosphere in Michigan, USA. PERMDISP with Tukey's HSD post-hoc test, * = $P$-value < 0.05, ** = $P$-value < 0.01. France $n$ = 5 plants per treatment; USA $n$ = 12 plants per treatment.

studies that attribute changes in rhizosphere microbial communities to plant stress (25, 27). Although Flavert's drought adaptation has not been studied, our data indicated a stronger negative effect of drought on this genotype, along with a possible legacy effect in the G2 rhizosphere microbiome, reflected also in the plant trait data. While it is possible that repeated drought exposure may drive plants to recruit drought-adapted microbes (62, 63), we did not find evidence of this in our study.

A key feature of our study was the full-factorial two-generation design using two common bean genotypes with distinct domestication histories. Flavert and Red Hawk genotypes originated from geographically isolated populations, with Red Hawk tracing its lineage to the Andean gene pool and Flavert being a European Flageolet bean (36, 53, 64). Bean genotype is known to influence rhizosphere microbiome structure and composition (41, 65), likely explaining the distinct microbiome responses observed between these two genotypes. Host genetics can shape plant microbial structure and composition (66–68), for example, by influencing root exudation patterns (69–71). Thus, genotype-driven exudate profiles can contribute to plant-microbe interactions under drought (72, 73), and it is possible that this mechanism may also be at play in the differences observed for these bean genotypes.

Red Hawk, a drought-susceptible red kidney bean (74), did not exhibit microbiome sensitivity despite its known drought susceptibility. This contrasts with studies reporting stronger microbiome shifts in drought-sensitive hosts (75, 76). While there may be some unknown experimental aspect that could act to stabilize the microbiome, what this

could be is not obvious. Furthermore, in earlier work, we found no drought effects on Red Hawk seed endophytes, although several bacterial taxa were stably transmitted across generations (31), suggesting no legacy effect of drought on the transmission of particular seed microbiome members. In addition, we previously observed only a weak effect of short-term drought on the active rhizosphere microbiome of Red Hawk despite visible plant stress (e.g., wilting and biomass reduction) (49). Similar observations suggest that microbial communities often show greater drought resistance than the plant itself (e.g., 77–79). However, there has been a report of rhizosphere microbiome adaptation to different magnitudes of drought (80). Thus, the observed microbiome stability in Red Hawk, despite plant declines, is notable and warrants further research to determine whether the microbiome resistance mitigates host stress.

Although previous studies have shown stronger drought effects on endophytes than on rhizosphere microbiomes (25, 81, 82), this pattern was not observed here. Root-associated (which included endophytes and rhizoplane together) and rhizosphere communities responded similarly to drought, except that Flavert's rhizosphere alpha diversity declined under G2 drought while root microbiomes remained stable.

We also detected differences in soil microbiome composition between locations, which likely reflect environmental and methodological variation between laboratories and include several factors that vary by location and thus cannot be partitioned (see Materials and Methods). Rather than being a limitation, this parallel, cross-continental design allowed direct comparison across distinct experimental contexts—an approach rarely reported but essential for identifying general patterns. Reporting each location separately would have obscured these valuable contrasts.

Finally, it may be tempting to add more analyses, for example, to report differentially abundant amplicon sequence variants and their taxonomies across the treatments. As the overarching trends show no notable differences across the treatments, this could be perceived as hunting for a more exciting outcome. It could be that considering the active component of the microbiome, rather than the "total" community inclusive of DNA from active, inactive, and relic taxa, would have yielded a different perspective on the community response to drought (49). We recommend future work to discriminate these components of the community, as the active taxa are those contributing to growth and plant-supportive functions.

In conclusion, our findings suggest that plant-associated microbiomes can remain stable (e.g., resistant) under drought stress despite host decline. Host genotype and soil origin modulated the microbiome responses, but we found no consistent evidence for a multi-generational drought legacy. Microbiome resistance, which may be underreported due to perceived "negative" results (55), is an important component of ecosystem stability. Understanding the mechanisms that reinforce stability, either by resistance or resilience, is critical for predicting plant-microbe responses to climate stress.

## MATERIALS AND METHODS

### Bean cultivars

*Phaseolus vulgaris* L. var. Red Hawk was selected as a dry bean variety (54) common in Michigan production and var. Flavert was selected as a Flageolet bean variety common in French production (53). Red Hawk seeds in Michigan were obtained from the Michigan State University Bean Breeding Program from its 2019 harvest and stored at 4°C until ready for use in experiments. Flavert and Red Hawk seeds in France were purchased from Vilmorin-Mikado (Limagrain group, France). These seeds were designated as G0 seeds and germinated to produce G1 plants.

### Field soil preparation

For G1 and G2 experiments performed in Michigan, agricultural field soil was collected from the Michigan State University Agronomy Farm, from a field that had grown common bean in 2019 (42°42'57.4"N, 84°27'58.9"W, East Lansing, MI, USA). Field soil was

used because it was expected to contain the legacy microbiome from the most recent bean crop and thus represent more authentically the microbiome and soil conditions for bean production. The soil was a sandy loam with an average pH of 7.2 and organic matter content of 1.9%, as assessed by the Michigan State University Soil Plant and Nutrient Laboratory by their standard protocols. The soil was collected before each planting group from the same field location. The 1–3 cm dry top layer of soil and plant debris was avoided. Soil was stored, covered, at 4°C until the experiment. Immediately before planting, the soil was passed through a 4 mm sieve to remove rocks and plant debris, and the soil was mixed with autoclaved coarse vermiculite at a 50% (vol/vol) ratio.

For G1 experiments performed in Pays de la Loire, the soil was collected from the experimental station of the National Federation of Seed Multipliers (FNAMS; 47°28′012.42″ N, 0°23′44.30″ W; Brain-sur-l'Authion, France), where common bean had been cultivated in 2016. This soil was a clay sand limestone with pH 7.1 and 1.9% organic matter content. The soil was sieved and mixed with coarse vermiculite using the abovementioned method. Soil for G2 was collected from the experimental fields belonging to the Institut National de Recherche pour l'Agriculture, l'Alimentation et l'Environnement (INRAE) in Angers, France (47°28′50.7″N, 0°36′31.4″W). A different soil was used in G2 to simulate a realistic scenario in which seed multiplication occurs at a site (seed producer) different from the bean production (farm). This soil had a sandy loam texture, with a pH of 6.5 and 1.9% organic matter content. The soil was sieved and mixed with vermiculite in the same fashion as G1.

## Surface sterilization and seed germination

For each Michigan planting group, Red Hawk seeds were surface-sterilized with a solution of 10% bleach and 0.1% Tween 20 before planting. Seeds were randomly selected from the G0 seed supply for G1 and from the harvested G1 seeds for G2. We avoided visibly cracked or moldy seeds. Seeds were placed in a Petri dish lined with sterile filter paper, and 1–2 mL sterile deionized (DI) $H_2O$ was applied to the filter paper. Petri dishes were stored in the dark at room temperature for 3–4 days for seeds to germinate, with an additional 2 mL sterile DI $H_2O$ added halfway through the germination period. Once seeds had sprouted radicle roots, they were transferred to the pots. Seeds planted in Pays de la Loire were germinated directly in the prepared pots of soil, without prior surface sterilization. The Pays de la Loire experiment included both Red Hawk and Flavert cultivars under the same soil and treatment conditions for each generation. After G1, seeds were removed from the pods of G1 plants, pooled by each plant in 50 mL conical tubes, and stored at 4°C until planting for the G2 experiment.

## Plant growth conditions

For the G1 experiment in Michigan, three germinated seeds were planted per 3.78 L (one gallon) pot and placed into a high-light BioChambers FLEX LED growth chamber with a 16-hour day/8-hour night cycle at 26°C and 22°C and 50% relative humidity. When seedlings emerged and reached the earliest vegetative stage ("VC," two cotyledons with primary leaves at nodes 1 and 2), they were thinned to one seedling per pot. Plants were watered every other day with 300 mL 0.05% 15N-10P-30K water-soluble fertilizer solution (Masterblend International, Morris, IL, USA). At the vegetative 3 stage ("V3," third trifoliate leaves expanded), stress treatments began for the drought-treated plants. Drought plants received 100 mL of 0.15% 15N-10P-30K fertilizer solution every other day (66% less water than control but the same amount of nutrients). After approximately 14 days of treatment, when plants reached the reproductive 1 stage ("R1," first open flowers), they were returned to the control watering every other day until senescence. There were 17 replicate plants grown per treatment in G1. Five plants per treatment were destructively harvested for plant phenotypic measurements at the reproductive 6 stage ("R6," most pods at the seed filling stage). The remaining 12 plants were grown until senescence (S). Mature seeds were collected from the 12 senesced plants for sowing

in the G2 experiment, and the root and rhizosphere microbiome were sampled from a subset of five replicate plants.

For the G2 experiment, seeds from 24 total G1 parental lines that received either control (12 lines) or drought (12 lines) were planted. There were four G1–G2 combinations: Control_Control, Control_Drought, Drought_Control, and Drought_Drought. Seeds from each parental line were evenly distributed among the Control and Drought G2 conditions and tracked to assess the influence of parent plant on the results. Seeds from each parental line were surface-sterilized and germinated as described above, then planted in the field soil-vermiculite mixture (from the same soil location) in seedling trays, and placed in the growth chamber under the abovementioned conditions. When plants reached the VC stage, four viable seedlings per parent line were transferred to 3.78 L (1 gallon) pots for the remainder of the experiment. Thus, each parental line had two offspring directed into each of the G2 conditions. One offspring was used for plant phenotypic measurements (12 plants total per G2 condition), while the other was harvested for microbiome analysis (12 plants total per G2 condition). Plants were watered according to the timeline described for G1.

Pays de la Loire plants were grown under the same growth chamber, control, and drought conditions described above. The Pays de la Loire experiment was replicated with Red Hawk and Flavert seeds. After three weeks of growth (day 18 after sowing, V3 stage), replicate plants ($n= 5$) were exposed to control conditions (300 mL of 0.05% nutritive solution) or drought (66% water withholding, 100 mL of 0.15% nutritive solution) for four weeks, until day 56, within the R5. This represented a slightly longer stress period than that applied to the plants in Michigan. Five plant parental lines from each G1 condition (10 lines total) from both genotypes were planted in G2. They were also planted in a complete factorial design with four treatment conditions in G2.

## Plant phenotypic data

Plants dedicated to phenotypic trait measurements at Michigan State were collected with a LI-COR LI-6800 to measure photosynthetic rate and stomatal conductance on the day before the drought ended within stage R1 (LI-COR Biosciences, Lincoln, NE, USA). Plants were grown until they reached approximately the reproductive 6 stage ("R6," pods developed with discernible green seeds that were not yet dry). Pods were removed and placed in a paper bag, plants were cut at the base of the stem, and shoots were placed in a large envelope. Roots were removed from the pot, shaken to remove excess soil, collected, rinsed, and put in a paper bag. Pods and seeds per plant were counted, and then, all three compartments were placed in a 50°C drying oven for two days. After drying, the shoot, root, and pod dry biomasses were measured.

In Pays de la Loire, plants were harvested for trait measurements at approximately the R5 stage. The entire root system was gently separated from the soil in the pot and placed in a plastic bag. The total fresh weight of each plant (both above- and below-ground tissues) was measured, and the number of pods and seeds per plant was counted.

## Microbiome compartment harvest

Once the Michigan plants had senesced and the pods were dried, the plants were harvested for microbiome analysis. Seed pods were removed and stored in a sterile Whirl-pak bag. Root systems were removed from the pot and shaken to remove loose soil. Roots were collected in a Whirl-pak bag, and associated rhizosphere soil was collected in a sterile 50 mL conical tube. Roots and rhizosphere soil were stored at −80°C until nucleic acids extraction.

The microbiome analysis in Pays de la Loire was performed on plants that were harvested at approximately the R5 stage. The entire root system was gently separated from the soil in the pot and placed in a plastic bag. The rhizosphere soil was collected by shaking the root system in a plastic bag and then stored at −80°C. The root system was washed in sterile distilled water, transferred to 50 mL tubes, and stored at −80°C.

## DNA extractions

The Michigan root samples were thawed at room temperature, and a 3–6 cm section of the main root system was cut and used for root DNA extraction, which combined the rhizoplane and endophytic bacteria of the root tissues. The selected sections were rinsed with sterile DI water and finely ground in liquid nitrogen with a mortar and pestle. DNA was extracted from the ground root material with the DNeasy PowerSoil Pro DNA Kit (Qiagen, Germantown, MD, USA) following the manufacturer's instructions, with the following modifications. In step one, 750 µL solution CD1 was used with 50 µL ATL buffer (Qiagen, Germantown, MD, USA). The bead beating step was performed for 15 minutes on a vortex genie 24-tube adapter at maximum speed. Lastly, 60 µL of the final elution buffer C6 was used, and tubes were incubated for 10 min before centrifugation.

Rhizosphere soil was thawed at room temperature, and DNA was extracted using the DNeasy PowerSoil Pro DNA Kit with the same ATL buffer modification as described for the roots. Negative controls (extraction reagent blanks) were included with each batch of DNA extractions, and one positive mock community control was included with each compartment sample set (roots or rhizosphere) (83). Controls were sequenced alongside the experimental samples.

French samples were processed similarly and extracted using the DNeasy Power-Soil Kit (Qiagen, Germantown, MD, USA, *discontinued*) following the manufacturer's instructions. A blank extraction kit control, a PCR-negative control, and a PCR-positive control (*Lactococcus piscium* DSM 6634, a fish pathogen that is not plant-associated) were included in each PCR plate.

## Sequencing

The 16S V4 rRNA gene was sequenced for the Michigan root and rhizosphere samples at the Argonne National Laboratory Environmental Sample Preparation and Sequencing Facility (Lemont, IL, USA). The DNA was PCR amplified with region-specific primers that included sequencer adapter sequences used in the Illumina MiSeq: FWD: GTGYCAGCMGCCGCGGTAA; REV: GGACTACNVGGGTWTCTAAT (84–88). Each 25 µL PCR reaction contained 9.5 µL of MO BIO PCR Water (Certified DNA-Free), 12.5 µL of QuantaBio's AccuStart II PCR ToughMix (2× concentration, 1× final), 1 µL Golay barcode tagged Forward Primer (5 µM concentration, 200 pM final), 1 µL Reverse Primer (5 µM concentration, 200 pM final), and 1 µL of template DNA. The conditions for PCR were as follows: 94°C for 3 min to denature the DNA, with 35 cycles at 94°C for 45 s, 50°C for 60 s, and 72°C for 90 s, with a final extension of 10 min at 72°C to ensure complete amplification. Amplicons were then quantified using PicoGreen (Invitrogen) and a plate reader (InfiniteÒ 200 PRO, Tecan). Once quantified, the volumes of each of the products were pooled into a single tube in equimolar amounts. This pool was then cleaned using AMPure XP Beads (Beckman Coulter) and quantified using a fluorometer (Qubit, Invitrogen). After quantification, the pool was diluted to 2 nM, denatured, and then diluted to a final concentration of 6.75 pM with a 10% PhiX spike for sequencing. Amplicons were sequenced on a 251 bp × 12 bp × 251 bp MiSeq run using customized sequencing primers and procedures (86).

For the 16S V4 rRNA gene sequencing in Pays de la Loire, PCR reactions were performed with a high-fidelity Taq DNA polymerase (AccuPrimeTM Taq DNA Polymerase System, Invitrogen) using 5 µL of 10× buffer, 1 µL of forward and reverse primers (10 µM), 0.2 µL of Taq, and 10 µL of DNA. A first PCR amplification was performed with the primer sets V4 515f/806r (5′-GTGCCAGCMGCCGCGGTAA-3′and 5′-GGACTACHVGGGTWTCTAAT-3′ (85). Cycling conditions consisted of an initial denaturation at 94°C for 3 min, followed by 35 cycles of denaturation at 94°C (30 s), primer annealing at 55°C (45 s), and extension at 68°C (90 s), with a final elongation at 68°C for 10 min. Amplicon purification was performed with a ratio of 0.8 of magnetic beads (Sera-MagTM, Merck). A second PCR amplification was performed to incorporate Illumina adapters and barcodes: a first denaturation at 94°C (1 min), followed by 12 cycles of denaturation at 94°C (60 s), primer annealing at 55°C (60 s), and extension at 68°C (60 s) with a final elongation at 68°C for

10 min. Amplicons were purified with a ratio of 0.7 of magnetic beads and quantified with the Quant-iTTM PicoGreen dsDNA Assay Kit (Invitrogen). All the amplicons were pooled in equimolar concentrations, and the concentration of the equimolar pool was monitored by quantitative PCR (KAPA SYBR FAST, Merck). Amplicon libraries were mixed with 10% PhiX and sequenced using a MiSeq Reagent Kit v3, 600 cycles (Illumina).

## Sequence data processing

Fastq files were processed in QIIME2 (v.2022.8.0) after primer removal (89) and demultiplexed with the demux emp-paired protocol. Samples were denoised, truncated, and merged at 100% sequence identity using DADA2 (90) in QIIME2, with the truncation lengths found in Table S1. 16S rRNA gene taxonomy was assigned using the SILVA database (91), release 132 for French data sets and release 138 for Michigan data sets, and taxonomy and amplicon sequence variant (ASV) tables were exported for further analysis in R.

## Plant phenotypic and microbiome data analysis

Data analyses were performed in R v.4.4.1 (92) and R Studio v.2025.5.1.513 (93). Amplicon sequence variants, taxonomy, and metadata tables were imported into the phyloseq package v.1.48.0 (94). Sequences derived from 16S rRNA genes unclassified at the phylum level or affiliated with Chloroplasts or Mitochondria were removed. The identification of sequence contaminants was assessed with the decontam package v.1.20.0 (95) using the prevalence of ASVs in samples and negative controls (Table S1). Rarefaction curves for each data set were generated using the rarecurve() function in the vegan package v.2.6.4 (96), and multiple rarefactions (without replacement, iterations of 100) were performed for each data sets using the phyloseq_mult_raref () function in the metagMisc package v.0.5.0 (97). The average of observed richness was calculated across multiple rarefactions using the phyloseq_mult_raref_div() function in the metagMisc package. The effects of drought treatment (G1 and G2) and plant genotype, as well as their interactions on richness, were assessed with Welch's two-sample $t$-test. The effect of drought treatment, plant genotype, and their interaction was assessed by two-way ANOVA. In addition, a three-way ANOVA was performed to analyze the effect of drought, genotype, and plant generation. Tukey's Honest Significant Difference (Tukey's HSD) post-hoc test was performed where applicable. The normality of residuals and homogeneity of the residual variances were verified using Shapiro-Wilk and Levene's tests, respectively. Response variable data were transformed, when necessary, using arcsinh, Box-Cox, or ordered quantile normalization (orderNorm) implemented in the bestNormalize package v.1.9.1 (98, 99). Microbiome alpha diversity figures were created using the ggplot2 package v.3.5.2 (100).

Bray-Curtis distances were calculated and averaged across rarefactions using the mult_dissim() and mult_dist_average() functions in the MetagMisc package. Permutational multivariate analysis of variance (PERMANOVA) was performed with the adonis2() function with 9,999 permutations from the vegan package to assess the effect of drought treatment, plant genotype, and their interactions on microbiome community composition. Due to the significant effect of planting group in the G2 Michigan data, we restricted the permutation by planting group using the setBlock() function in the permute package v.0.9.8 (101). Post-hoc analysis of the PERMANOVA results was performed with the pairwise.adonis2() function from the pairwiseAdonis package v.0.4.1 (102). Principal coordinates analysis (PCoA) plots were created from Bray-Curtis dissimilarities using the cmdscale() function in the stats package v.4.4.1, with ggplot2 used for visualization. For the G2 Michigan data, we performed partial constrained analysis of principal coordinates using capscale() with the Condition() function in the vegan package to control the effect of planting group. Beta dispersion was assessed using the betadisper() and permutest() functions from the vegan package with the spatial median method (103).

The effect of drought, plant genotype, and their interactions on plant phenotypes was analyzed with Welch's two-sample *t*-test, two-way ANOVA, or three-way ANOVA, with Tukey's HSD post-hoc tests where applicable. The data normality and homogeneity were assessed using Shapiro-Wilk and Levene's tests, respectively. Response variable data were arcsinh or square-root transformed as needed using the bestNormalize package. Plant phenotype figures were created with ggplot2. Additional data processing and statistics were performed in the tidyverse package v.2.0.0 (104) and R stats. Figure panels were assembled with the patchwork package v.1.3.0 (105).

## ACKNOWLEDGMENTS

This work was co-funded by the United States Department of Agriculture (grant number 2019-67019-29305) to A.S. and M.B. and by the Michigan State University Plant Resilience Institute to A.S. This work was also co-funded by the European Union [grant number ERC, MicroRescue, 101087042] to A.S. Views and opinions expressed are however those of the author(s) only and do not necessarily reflect those of the European Union or the European Research Council. Neither the European Union nor the granting authority can be held responsible for them.

A.S. acknowledges support from the United States Department of Agriculture National Institute of Food and Agriculture and Michigan State University AgBioResearch and the Center National de la Recherche Scientifique (CNRS), France. The Angers Plant Phenotyping Platform PHENOTIC (DOI: 10.17,180/ykbz-2v85) is acknowledged for the production and phenotyping of plants.

## AUTHOR AFFILIATIONS

[1]Laboratoire d'Ecologie Microbienne LEM, Universite Claude Bernard Lyon 1, CNRS, INRAE, VetAgro Sup, CNRS UMR5557, INRAE UMR1418, Villeurbanne, France

[2]Department of Microbiology, Genetics and Immunology; Program in Ecology, Evolution and Behavior, Plant Resilience Institute, Michigan State University, East Lansing, Michigan, USA

[3]Univ Angers, Institute Agro, INRAE, IRHS, SFR QUASAV, Angers, France

## AUTHOR ORCIDs

A. Fina Bintarti  http://orcid.org/0000-0002-5913-3081
Abby Sulesky-Grieb  https://orcid.org/0000-0001-6505-5617
Marie Simonin  http://orcid.org/0000-0003-1493-881X
Matthieu Barret  http://orcid.org/0000-0002-7633-8476
Ashley Shade  http://orcid.org/0000-0002-7189-3067

## FUNDING

| Funder | Grant(s) | Author(s) |
| --- | --- | --- |
| U.S. Department of Agriculture | 2019-67019-29305 | Matthieu Barret |
| | | Ashley Shade |
| Michigan State University Plant Resilience Institute | | Ashley Shade |
| European Research Council | MicroRescue 101087042 | Ashley Shade |

## AUTHOR CONTRIBUTIONS

A. Fina Bintarti, Data curation, Formal analysis, Investigation, Visualization, Writing – original draft, Writing – review and editing | Abby Sulesky-Grieb, Data curation, Investigation, Visualization, Writing – original draft | Joanna Colovas, Data curation, Investigation, Writing – review and editing | Brice Marolleau, Data curation, Investigation, Writing – review and editing | Tristan Boureau, Data curation, Investigation,

Writing – review and editing | Marie Simonin, Data curation, Formal analysis, Investigation, Visualization, Writing – review and editing | Matthieu Barret, Conceptualization, Data curation, Formal analysis, Funding acquisition, Investigation, Project administration, Supervision, Writing – review and editing | Ashley Shade, Conceptualization, Formal analysis, Funding acquisition, Investigation, Project administration, Supervision, Visualization, Writing – original draft, Writing – review and editing

## DATA AVAILABILITY

Data analysis code can be found at (https://github.com/ShadeLab/Drought_multigeneration_study_common_bean). Raw sequence data from the French samples can be found on the European Nucleotide Archive under accession number PRJEB65346. Raw sequences from the Michigan samples can be found on the NCBI Sequence Read Archive under BioProject accession number PRJNA1058980.

## ADDITIONAL FILES

The following material is available online.

### Supplemental Material

**Supplemental material (Spectrum03019-25-s0001.pdf).** Fig. S1; Tables S1 to S14.

### Open Peer Review

**PEER REVIEW HISTORY (review-history.pdf).** An accounting of the reviewer comments and feedback.

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
