## [Reviewer comments · Microbiology Spectrum]

Microbiology Spectrum

Evaluating the legacy of drought exposure on root and rhizosphere bacterial microbiomes over two plant generations

A. Bintarti, Abby Sulesky-Grieb, Joanna Colovas, Brice Marolleau, Tristan Boureau, Marie SIMONIN, Matthieu BARRET, and Ashley Shade

Corresponding Author(s): Ashley Shade, CNRS Delegation Alpes

Review Timeline:

Submission Date:	September 22, 2025
Editorial Decision:	October 27, 2025
Revision Received:	October 29, 2025
Editorial Decision:	December 16, 2025
Revision Received:	January 16, 2026
Accepted:	February 11, 2026

Editor: Weimin Sun

Reviewer(s): Disclosure of reviewer identity is with reference to reviewer comments included in decision letter(s). The following individuals involved in review of your submission have agreed to reveal their identity: Aqleem Abbas (Reviewer #2)

Transaction Report:

DOI: <https://doi.org/10.1128/spectrum.03019-25>

Re: Spectrum03019-25 (**Evaluating the legacy of drought exposure on root and rhizosphere bacterial microbiomes over two plant generations**)

Dear Prof. Ashley Shade:

Thank you for the privilege of reviewing your work. Below you will find my comments, instructions from the Spectrum editorial office, and the reviewer comments.

Revision Guidelines

Sincerely,
Weimin Sun
Editor
Microbiology Spectrum

Reviewer #1 (Comments for the Author):

In addition to management strategies, the detrimental effects of drought on agricultural yields can be offset by the actions of mycorrhizal fungi and bacteria in the rhizospheres of crops. These beneficial organisms express a wide range of functional traits to help plants to cope with drought and promote crop acclimatation to drought conditions. An important question, supported by some evidence, is whether one can enrich the populations of these beneficial organisms by exerting abiotic or biotic stress on

the system. In this study, Bintarti et al. conducted pot experiments to investigate the long-term effects of repeated droughts on the root and rhizosphere microbiome of the common bean. This is a common occurrence in many environments and across plant generations it is also linked to the visioning of climate resilient crop rotation. This setup was more realistic than the single drought treatment often employed, which involves ceasing to water the plants. It called into question the resilience of the microbiome, which could then contribute to the resilience of crops subjected to drought at the next incidence of drought. Experimental design is well described. Sampling rhizospheres directly after the drought and working at the RNA level instead of the DNA level would have provided more information about the microbiome's response to the drought, as with the DNA level the drought response takes longer to establish and may not be visible if drought is short. The reasoning of using DNA and the sampling design should be elaborated. The ASVs were not affiliated to taxa but instead used as a diversity measure.

In generation 1, the effects in France, where the stress period was slightly longer, were stronger than in the US. Three weeks after the drought treatment, negative effects on plant photosynthesis, pod production and above-ground biomass were observed, which suggested that the treatment had worked. In the US, however, the effects were more subtle. Root biomass was not greatly affected, although a shorter drought period often results in a higher root-to-shoot ratio. Unfortunately, root architecture or root exudation was not assessed. Second-generation plants responded very strongly to drought; however, this response was not related to drought legacy. This was surprising. This time, the root system of the Red Hawk genotype responded too, producing more root biomass in France. Once again, the results of the US experiment differed from those in France. The rhizosphere microbiome did not respond strongly to drought, suggesting that root exudation was unaffected by this level of drought over this period, either because there was low impact on rhizodeposition, or because the composition returned to normal following later watering during plant development (recovery). Dispersion analysis supported the finding that beta diversity changed for the Flavert genotype, but not the Red Hawk genotype. In G2, Flavert root beta diversity was affected by the interaction between the G1 and G2 treatments, meaning the effect of drought on G2 was influenced by G1. Dispersion from C to D was higher than from D to C in the Red Hawk root and rhizosphere in G2, suggesting recovery from drought in the latter. Although this theoretical perspective is well described, it should be accompanied by the reporting of the differentially abundant ASVs and their taxonomies across the treatments. While the manuscript is interesting and well written, it is lacking root exudate and ASV data. However, as the ASV data exist, I would suggest to show that - focusing specifically to the drought, drought legacy - and their relation to plant genotype.

Reviewer #2 (Comments for the Author):

Dear Authors

I have read the manuscript throughly, however, usage of legacy term through the manuscript is little bit awkward, when I search the definition, i find this definition [enduring or residual effects of drought exposure that persist in the root and rhizosphere bacterial microbiomes, influencing their composition, function, or resilience across subsequent plant generations, even after the drought stress has ended.] Very lengthy definition, before proper revision, I request to modify legacy terms and use already present terms, your work should be novel not novel terms.

Reviewer #4 (Comments for the Author):

The manuscript attempted to explore the legacy of serial drought exposures on root endophytic and rhizospheric bacterial communities between two generations if *Phaseolus vulgaris*.

The experiment is well designed. However, the primary problem is the manuscript did not properly interpret the following questions before authors started their research: how legacy effect affects the growth performances and stress resistance of plants, how it can be identified, and how it can be assessed with the approaches relied on alpha-, beta-diversity data of phytobiomes. These questions should be well discussed in the Introduction, and be responded again in the Discussion, focusing on the results yielded in the research. Unfortunately, the manuscript did not complete such work in my view.

The writings in Methods and Materials is better than those in Introduction, but the analysis of microbiome is too limited: it only compares alpha- and beta-diversity, lacks details such as microbial taxonomic composition, and fails to fully exploit the information generated by the methodology.

The discussion fails to focus on the key scientific questions and is not sufficiently concise. For example, readers cannot conclude the causes, mechanisms of effects, influential factors, and outcomes of the legacy effect in *Phaseolus vulgaris* from the discussion.

In summary, the manuscript presents the research and results in an ambiguous way and thus needs to improve the strategies of writing and data analysis.

Minor comments.

Line 37: please consider delete ", while difficult to implement,"

Line 45: consider change "standard for" to "routine in"

Line 56: please consider modify this sentence, for example, "They collectively provide vital ecosystem services to agriculture,

including improving water absorption and nutrient assimilation by the plant, nutrient cycling in the soil and root system, and enhancing resistance to pathogens".

There are many sentences that are grammatically correct but use awkward terminology and collocations throughout the manuscript. I regret to say I am unable to edit all of them individually. A comprehensive language edit by a native English speaker is recommended.

Line 60-64: empty, please be more specific and depict the research progress to the issue.

Line 71: please consider delete "Our study aimed to understand the implications of repeated drought on the microbiome of the legume common bean (*Phaseolus vulgaris* L.)." here as this paragraph has not reached your research yet.

Line 87-93: empty. What facts have those researches reported, what pattern have they revealed, and what questions has been left to be solved by the present study?

Line 98: Why is hypothesis (2) proposed? With no preceding groundwork in the text, its introduction here appears baseless

Line 107-113: this might be good to be placed in the Methods and Materials.

Line 125: Pays de la Loire experiments used soil from two sites for G1 and G2, respectively. Was the soil from Michigan replaced with new soil in G2 experiment, or did the G1 and G2 plants grow in the same soil intended to assess the plants and microbiome under continuous cultivation?

Line 469-470: Usually this would be present in Discussion.

Line 568-571: Taxonomic features of microbiome are worthy to be demonstrated in the manuscript in my view.

Line 852: "in G2"

Thank you for the three reviews. In our response to reviews:

- We offer our responses preceded by >>> and in **bold font**
- We have provided manuscript line numbers that align to the Tracked Changes document (and not to the revised “clean” main text).
- To organize our response, we have numbered each specific comment and used the comment tool in the Tracked Changes to link the numbered comments to the revisions made.
- We have used the yellow highlight to emphasize the reviewer question/comment to be addressed.
- In some instances, we have used the red to improve the grammar or clarity of the reviewer text to show how we understood and interpreted the comment.

Spectrum03019-25 (Evaluating the legacy of drought exposure on root and rhizosphere bacterial microbiomes over two plant generations)

Reviewer #1 (Comments for the Author):

R1.1

In addition to management strategies, the detrimental effects of drought on agricultural yields can be offset by the actions of mycorrhizal fungi and bacteria in the rhizospheres of crops. These beneficial organisms express a wide range of functional traits to help plants to cope with drought and promote crop acclimatation to drought conditions. An important question, supported by some evidence, is whether one can enrich the populations of these beneficial organisms by exerting abiotic or biotic stress on the system. In this study, Bintarti et al. conducted pot experiments to investigate the long-term effects of repeated droughts on the root and rhizosphere microbiome of the common bean. This is a common occurrence in many environments and across plant generations it is also linked to the visioning of climate resilient crop rotation. This setup was more realistic than the single drought treatment often employed, which involves ceasing to water the plants. It called into question the resilience of the microbiome, which could then contribute to the resilience of crops subjected to drought at the next incidence of drought. Experimental design is well described. Sampling rhizospheres directly after the drought and working at the RNA level instead of the DNA level would have provided more information about the microbiome's response to the drought, as with the DNA level the drought response takes longer to establish and may not be visible if drought is short. The reasoning of using DNA and the sampling design should be elaborated. The ASVs were not affiliated to taxa but instead used as a diversity measure.

>>>Thank you for the review of the manuscript and for the positive comments about the scope and experimental design.

>>> We agree that, for a short-term response to drought, considering the change in activity state using RNA-based or other activity discriminating methods can be useful. Indeed, we have published work concerning discriminating the active component in plant microbiomes in response to short-term stress (e.g., Bandopadhyay et al. 2024, Bandopadhyay et al. 2025). However, in the present study we were interested in the legacy of drought on the selected and potentially transmitted microbiome members given multiple generational plant exposure, and thus the selection of communities and their taxa by the conditions and host plant to this stress. Therefore, DNA based assays, which reveal “who is there” are appropriate for questions of long term (e.g. multi-generational) selection of communities and taxa.

We also have noted in the discussion that assessing activation of detected could be future work:

“It could be that considering the active component of the microbiome, rather than the “total”, inclusive of DNA from active, inactive, and relic taxa, would have yielded a different perspective on the community response to drought (Bandopadhyay *et al.* 2024). We recommend future work to discriminate these components of the community, as the active taxa are those contributing to growth and plant-supportive functions.”

R1.2

In generation 1, the effects in France, where the stress period was slightly longer, were stronger than in the US. Three weeks after the drought treatment, negative effects on plant photosynthesis, pod production and above-ground biomass were observed, which suggested that the treatment had worked. In the US, however, the effects were more subtle. Root biomass was not greatly affected, although a shorter drought period often results in a higher root-to-shoot ratio. Unfortunately, root architecture or root exudation was not assessed. Second-generation plants responded very strongly to drought; however, this response was not related to drought legacy. This was surprising. This time, the root system of the Red Hawk genotype responded too, producing more root biomass in France. Once again, the results of the US experiment differed from those in France. The rhizosphere microbiome did not respond strongly to drought, suggesting that root exudation was unaffected by this level of drought over this period, either because there was low impact on rhizodeposition, or because the composition returned to normal following later watering during plant development (recovery). Dispersion analysis supported the finding that beta diversity changed for the Flavert genotype, but not the Red Hawk genotype. In G2, Flavert root beta diversity was affected by the interaction between the G1 and G2 treatments, meaning the effect of drought on G2 was influenced by G1. Dispersion from C to D was higher than from D to C in the Red Hawk root and rhizosphere in G2, suggesting recovery from drought in

the latter. Although this theoretical perspective is well described, it should be accompanied by the reporting of the differentially abundant ASVs and their taxonomies across the treatments. While the manuscript is interesting and well written, it is lacking root exudate and ASV data. However, as the ASV data exist, I would suggest to show that - focusing specifically to the drought, drought legacy - and their relation to plant genotype.

>>>Thank you for this note, and for the positive comment that the manuscript is interesting and well written. For the comment about the differentially abundant ASVs given the treatment and why we have not performed this analysis, this was a careful consideration, which we address this note directly in the discussion (~lines 568-575):

“Finally, it may be tempting to add more analyses, for example, to report differentially abundant ASVs and their taxonomies across the treatments. As the overarching trends show no notable differences across the treatments, and this could be perceived as hunting for a more exciting outcome.”

>>>In brief, the few observed treatment effects were very weak and inconsistent, and so we do not wish to overextend or overinterpret these data given that reality.

Reviewer #2 (Comments for the Author):

R2.1

Dear Authors

I have read the manuscript thoroughly, however, usage of legacy term through the manuscript is little bit awkward, when I search the definition, i find this definition [enduring or residual effects of drought exposure that persist in the root and rhizosphere bacterial microbiomes, influencing their composition, function, or resilience across subsequent plant generations, even after the drought stress has ended.] Very lengthy definition, before proper revision, I request to modify legacy terms and use already present terms, your work should be novel not novel terms.

>>>Thank you for your thorough read of the manuscript and comment about the term legacy. We have now added the definition to the introduction. (L72)

Reviewer #4 (Comments for the Author):

The manuscript attempted to explore the legacy of serial drought exposures on root endophytic and rhizospheric bacterial communities between two generations of *Phaseolus vulgaris*.

R4.1

The experiment is well designed. However, the primary problem is **that** the manuscript did not properly interpret the following questions before authors started their research: how legacy effect affects the growth performances and stress resistance of plants, how it can be identified, and how it can be assessed with the approaches relied on alpha-, beta-diversity data of phytobiomes. These questions should be well discussed in the Introduction, and be responded again in the Discussion, focusing on the results yielded in the research. Unfortunately, the manuscript did not complete such work in my view.

>>>Thank you for reviewing the work, for the positive comments on the design of the experiment, and for the helpful suggestions.

>>>We interpret this comment to mean that the reviewer would like more clarity as far as the scientific questions and approaches regarding the legacy of drought. At the request of R2, we have now added a definition of legacy to the introduction, which we believe clarifies this. We also have added clarifying text to the existing paragraph in the introduction that discussions how drought effects legumes and common bean, including our referenced work of investigating legacy of drought on the seed endophyte microbiome. As we are investigating the legacy of the drought on the microbiome itself, and so standard metrics of alpha and beta-diversity are used to assess community-level responses to stress (e.g., changes in the number of detected taxa and changes in the dynamics of community structure). We have already discussed the legacy of drought (lack of response) throughout the Results (e.g., see the paragraph starting “Inconsistent drought and drought legacy effects...”) and Discussion (paragraphs 1-6 therein)

R4.2

The writings in Methods and Materials ~~is~~ **are** better than those in Introduction, but the analyses of microbiome ~~is~~ **are** too limited: it only compares alpha- and beta-diversity, **and** lacks details such as microbial taxonomic composition, and fails to fully exploit the information generated by the methodology.

>>> Thank you for this comment, which is like R1.2. It is true that we do not report the differential abundances of taxa across the treatments. As we describe in the Discussion, this was a careful, intentional choice that reflects the fact that there were no treatment effects and inconsistent legacy effects detected. Therefore, to report differentially abundant taxa across treatments that had no effects would be to somehow suggest that there was a treatment effect, which we feel would be an overinterpretation or even misleading given the data. Here is our justification from the discussion (~lines 568-575):

“Finally, it may be tempting to add more analyses, for example, to report differentially abundant ASVs and their taxonomies across the treatments. As the overarching trends show no notable differences across the treatments, and this could be perceived as hunting for a more exciting outcome.”

R4.3

The discussion fails to focus on the key scientific questions and is not sufficiently concise. For example, readers cannot conclude the causes, mechanisms of effects, influential factors, and outcomes of the legacy effect in *Phaseolus vulgaris* from the discussion.

In summary, the manuscript presents the research and results in an ambiguous way and thus needs to improve the strategies of writing and data analysis.

>>>This comment is unclear, and so we are not sure what is required in a revision. As we did not observe a legacy effect of drought on the microbiome given multiple generations of drought exposure in our experiment, and so we cannot comment directly from these data on their mechanisms, influential factors, or outcomes. However, we do discuss the factors that likely played a role in the differences across genotypes and locations in the Discussion, which reflect the major results.

>>>We have made general edits to the Discussion to improve conciseness, specifically by removing redundancies while improving flow.

Minor comments.

R4.4

Line 37: please consider deleting ", while difficult to implement,"

>>>We have deleted this phrase

R4.5

Line 45: consider changing "standard for" to "routine in"

>>>We have rephrased the sentence for clarity.

R4.6

Line 56: please consider modifying this sentence, for example, "They collectively provide vital ecosystem services to agriculture, including improving water absorption and nutrient assimilation by the plant, nutrient cycling in the soil and root system, and enhancing resistance to pathogens".

There are many sentences that are grammatically correct but use awkward terminology and collocations throughout the manuscript. I regret to say I am unable to edit all of them individually. A comprehensive language edit by a native English speaker is recommended.

>>>Thank you, we have broken down this sentence to be clearer.

>>> The corresponding author and co-first author are native English speakers. The first draft of this manuscript was prepared by these authors, and the final draft was comprehensively edited for language and style by the corresponding author. In addition, we used the paid subscription service "Grammarly" to improve the grammar, flow, and wording of the

manuscript. While we appreciate the opportunity to improve the clarity of the writing, we gently and politely recommend that the reviewer tries not to make assumptions about the language capacities of authors based on their geographic locations, names, etc, which could be interpreted to reflect a negative bias of the reviewer against non-native speakers.

R4.7

Line 60-64: empty, please be more specific and depict the research progress to the issue.

>>> Thank you. We have improved this paragraph, which is intended to motivate the need to study repeated stress exposures for plant microbiomes generally and for drought specifically:

“While many studies have investigated the impacts of environmental stress on the plant microbiome (e.g., Santos-Medellín *et al.* 2017; Mavrodi *et al.* 2018; Timm *et al.* 2018; Veach *et al.* 2020; Wang *et al.* 2020; Russell and McFrederick 2022; Tiziani *et al.* 2022), very few have studied the effects of repeated stress and over multiple generations (Rodríguez *et al.* 2023; Sulesky-Grieb *et al.* 2024). Repeated stress exposure that occurs over multiple growing seasons and plant generations is expected to have negative impacts on crop agriculture, for example, by reducing seed quality and degrading soil quality (Furtak and Wolińska 2023; Maity *et al.* 2023). Compounded drought stress is of particular concern with climate change, as repeated exposure to drought is expected to become more frequent for many production areas.”

R.4.8

Line 71: please consider delete "Our study aimed to understand the implications of repeated drought on the microbiome of the legume common bean (*Phaseolus vulgaris* L.)" here as this paragraph has not reached your research yet.

>>>Unfortunately, this comment is not clear. This sentence is an accurate statement of our study objective and is aligned to our results, and so we do not see the reason to delete it. Please advise.

R4.9

Line 87-93: empty. What facts have those ~~researches~~ **studies** reported, what pattern have they revealed, and what questions has been left to be solved by the present study?

>>>We interpret this comment to mean that the reviewer would like to see added some more specific statements of our previous results to this section. Therefore, we have expanded here.

R4.10

Line 98: Why is hypothesis (2) proposed? With no preceding groundwork in the text, its introduction here appears baseless

>>>Thank you for this comment. We have added two references regarding expected microbiome responses to disturbances (Anna Karenina hypothesis, as also discussed in the beta-dispersion results section) that motivate this hypothesis.

- Jurburg, S.D., Blowes, S.A., Shade, A., Eisenhauer, N. and Chase, J.M., 2024. Synthesis of recovery patterns in microbial communities across environments. *Microbiome*, 12(1), p.79.
- Zaneveld, J.R., McMinds, R. and Vega Thurber, R., 2017. Stress and stability: applying the Anna Karenina principle to animal microbiomes. *Nature microbiology*, 2(9), pp.1-8.

R4.11

Line 107-113: this might be good to be placed in the Methods and Materials.

>>>Thank you, because the Journal formatting requires that Materials and Methods are provided after the Discussion, we have moved this paragraph from the end of the Introduction to the top of the Results section.

R4.12

Line 125: Pays de la Loire experiments used soil from two sites for G1 and G2, respectively. Was the soil from Michigan replaced with new soil in G2 experiment, or did the G1 and G2 plants grow in the same soil intended to assess the plants and microbiome under continuous cultivation?

>>>Thank you for this question. We used the same soil location for both G1 and G2 in Michigan, as now clarified on ~Line 225.

R4.13

Line 469-470: Usually this would be present in Discussion.

>>>This comment is related to comment R4.10. This sentence motivates the need to assess variance/dispersion based on the Anna Karenina hypothesis. As there was no treatment effect on dispersion, we did not see there was need to re-visit these results in the discussion.

R4.14

Line 568-571: Taxonomic features of microbiome are worthy to be demonstrated in the manuscript in my view.

>>>We interpret that this comment is redundant with the reviewer's previous comment R4.2, please see response to R4.2.

R4.15

Line 852: "in G2".

>>>Thank you for this comment. Consistent with all the figure legends, we used the full name, "Generation 2", instead of the abbreviation "G2" for clarity because this sentence is the title of the figure and we would like it to

be able to stand alone without the need for readers to search for the annotation elsewhere in the main text.

Re: Spectrum03019-25R1 (**Evaluating the legacy of drought exposure on root and rhizosphere bacterial microbiomes over two plant generations**)

Dear Prof. Ashley Shade:

Thank you for the privilege of reviewing your work. Below you will find my comments, instructions from the Spectrum editorial office, and the reviewer comments.

One of the reviewers was unable to complete the review due to time constraints. However, they did note that "the discussions are empty and lack theoretical depth," and suggested that the author should focus on improving this aspect in the revision.

Revision Guidelines

Sincerely,
Weimin Sun
Editor
Microbiology Spectrum

Reviewer #1 (Comments for the Author):

The authors responded appropriately to my suggestions and added some text as well. Although I think it would have been interesting to see the differentially abundant ASVs, I understand the reasoning behind not showing them - to avoid obscuring the main result.

Reviewer #2 (Comments for the Author):

Summary

The main findings of this research are that the microbiome is resistant primarily to drought despite clear host plant stress, which is essential and counter to many assumptions in the field. However, the manuscript has several weaknesses that need to be addressed to strengthen the interpretation of the results and bolster the claims.

1. Key differences between the two locations (soil source, harvest time, sterilization) confound the "location" effect. Please clarify this?
2. Resistant is frequently used in the manuscript; however, the data show no statistical effect, and it is not the same as proven stability.
3. Statistical reporting is inconsistent, and the figures should better represent the complex factorial design.
4. Explanation for the observed resistance, such as soil buffering, vermiculite use, has not been discussed.
5. Abstract: Too broad and overstated. Rhizosphere microbiomes can be resistant to drought stress". Only one example of a legacy flavor effect is mentioned, creating a slight imbalance. Therefore, there is a need to reemphasize the genotype-specific and context-dependent nature of response, for example, "resistance to drought in the Red Hawk genotype across soils, while the Flaverent genotype showed minor legacy effects
6. Introduction: I think there is a need to refine the hypothesis, to explicitly mention testing for both compositional shifts (beta diversity) and increased heterogeneity (beta dispersion) as indicators of disturbance, and to rewrite the introduction.
7. Results: The problem of using different soils in G1 and G2 for the French experiment is a major confounding factor. Any "legacy" effect in France could be masked or altered by the completely new soil microbiome in G2. This severely limits the interpretation of the French G2 legacy results. Moreover, Different harvest timepoints (R5 in France vs. Senescence in Michigan) indicate the microbiomes were sampled at different ecological successional stages. A drought effect at R5 might recover by senescence, making direct comparisons of "effect size" between locations invalid. Different seed sterilization protocols indicate the initial microbial inoculum for plants was fundamentally different, indicating location effects beyond just the soil.
8. Results. Often, interpret a non-significant PERMANOVA as "resistance." A non-significant result could be due to high variability, low statistical power, or a true lack of effect. The authors should say "we did not detect a significant effect of drought" rather than asserting "the microbiome was resistant." The beta-dispersion analysis helps here, but the language should be precise
9. I could not understand why some PERMANOVA R^2 values are very low (e.g., $R^2=0.03$ for Red Hawk G2 roots in France), even when significant ($P=0.029$). These should be highlighted as very weak effects, explaining almost none of the variance.
10. Michigan G2 rhizosphere result is confusing: "a global interaction... was initially detected... but post-hoc pairwise tests did not support it." This suggests the overall model may be overfitted or underpowered, and the result should be treated as non-significant.
11. Figures 5, 7, and 8 use color and shape to show a 2x2 factorial design (G1_G2), but it remains visually challenging to interpret. A different plot type (e.g., faceted bar plots for alpha diversity) might more clearly show the main and interaction effects.
12. Discussion: You have focused on the methodological differences as a strength for identifying generalities, but do not sufficiently acknowledge them as "limitations for mechanistic interpretation." For example usage of a 50/50 soil-vermiculite mix, while justified for drainage, creates an artificial, low-carbon environment that likely reduces microbial biomass and buffers against stress, potentially causing the "resistance" observed. Again, overinterpretation or overstating of stability. For example, "microbiome resistance is an important component of ecosystem stability" is a broad ecological claim. The research shows a lack of response in one system; it does not provide evidence that this resistance is a general or adaptive feature that contributes to ecosystem function.
13. I did not see the full report of the differential abundance analysis. While "fishing" is a concern, a standard, non-hypothesis-driven test for differentially abundant taxa (e.g., using DESeq2 or ANCOM-BC) is an expected part of microbiome papers.
14. I could not find the reasons for the critical method differences between locations (especially the different G2 soil in France and the different harvest timepoints). This sentence is good: "A different soil was used in G2 to simulate..." is good, but why wasn't this done for Michigan?

Minor comments

1. The research is not simply about microbiome resistance; it is actually about "Microbiome resistance to drought is genotype-specific and context-dependent, and multi-generational legacy effects are minimal or absent in the systems we tested. The manuscript should reflect this more nuanced conclusion.
2. Use consistent language throughout the manuscript ("no significant effect was detected") and evidence of absence ("the community was resistant"). Report effect sizes (R^2) alongside p-values will not help to judge the biological importance of statistical results.

Suggestions

1. To verify claims of functional stability, perform any of the resistance genes only on archived samples, to test whether the microbiome shows transcriptional resistance or activates another stress response pathway.
2. It might be that the resistance is an emergent property of native soil or an artificial soil plus vermiculite mixture. Please see this
3. If you have time, I suggest introducing a 3rd-generation challenge with a more severe drought to unmask any potential cryptic legacy effects on plant resilience that were not evident from compositional microbiome data alone. Please think about this

Thank you for the three reviews. In our response to reviews:

- We offer our responses preceded by >>> and in **bold font***
- We have provided manuscript line numbers that align to the Tracked Changes document (and not to the revised “clean” main text).*
- To organize our response, we have numbered each specific comment and used the comment tool in the Tracked Changes to link the numbered comments to the revisions made.*

Re: Spectrum03019-25R1 (**Evaluating the legacy of drought exposure on root and rhizosphere bacterial microbiomes over two plant generations**)

Dear Prof. Ashley Shade:

Thank you for the privilege of reviewing your work. Below you will find my comments, instructions from the Spectrum editorial office, and the reviewer comments.

R0

One of the reviewers was unable to complete the review due to time constraints. However, they did note that "the discussions are empty and lack theoretical depth," and suggested that the author should focus on improving this aspect in the revision.

>>>We thank the reviewer for taking the time to read the discussion of manuscript. It would be useful to have some specific suggestions as to which aspects of the discussion were found to be lacking. Nonetheless, we have added some theoretical context to the discussion by adding an explanation of the ecological concepts of resistance, resilience, and stability, which is also aligned with R2.2’s request.

Thus, the discussion currently includes five focused discussion points:

- 1. Discussion of ecological resistance (**NEW, see R2.2**) – its definition and usage in the experimental context**
- 2. Discussion of the resistance attributes of the microbiome specifically in the context of drought as referenced to the literature**
- 3. Discussion of the impact of bean plant genotype on microbiome and drought responses, including in exudation and microbiome recruitment as referenced to the literature**
- 4. Placement of our new results within the context of our previous work on the responses of plants and their microbiomes to drought, as referenced and compared to our previously published work in this area.**

5. Discussion of the limitations and strengths of the study, including cautions with over-interpretation of certain results and the merits of cross-experiment comparisons.

The discussion is well-referenced, with 30 citations therein that provide a foundation from the literature. Finally, we kindly note that we had made quite extensive improvements to the discussion in the previous revision based on comments from the previous reviewers to “focus on the key questions” and “make it more concise” (e.g., previous R4.3).

Reviewer #1 (Comments for the Author):

The authors responded appropriately to my suggestions and added some text as well. Although I think it would have been interesting to see the differentially abundant ASVs, I understand the reasoning behind not showing them - to avoid obscuring the main result.

>>>We thank the reviewer for taking the time to consider the revision and also are glad that they are satisfied with them.

Reviewer #2 (Comments for the Author):

Summary

The main findings of this research are that the microbiome is resistant primarily to drought despite clear host plant stress, which is essential and counter to many assumptions in the field. However, the manuscript has several weaknesses that need to be addressed to strengthen the interpretation of the results and bolster the claims.

>>>Thank you for taking the time to review the work and we're glad that you found it essential and interesting given assumptions made in the field.

R2.1. Key differences between the two locations (soil source, harvest time, sterilization) confound the "location" effect. Please clarify this?

>>>Thank you for this key comment. The location effect includes all these potential differences across locations, which are transparently disclosed in the methods and lumped together into the “site” factor for analysis. We have clarified this in the discussion:

“We also detected differences in soil microbiome composition between locations, which likely reflect environmental and methodological variation between laboratories and includes several factors that vary by location and thus cannot be partitioned (see Methods). “

R2.2. Resistant is frequently used in the manuscript; however, the data show no statistical effect, and it is not the same as proven stability.

>>>Thank you for this clarifying question about the relationship between resistance and stability. In brief, stability includes two components: resistance and resilience (e.g., Pimm 1984, Orwin and Wardle 2004, Shade et al. 2012). Resistance, the opposite of sensitivity, is the ability of a system to remain unchanged in the face of disturbance. Resilience is the capacity of a system to recover after being altered by a disturbance. If a system is unchanged when faced with a disturbance, it is not possible to observe resilience. Therefore, in the absence of change, resistance remains the sole component of stability. In other words, the absence of a response is equivalent to a resistant, and thus a stable, system.

Relevant references:

- Shade A, Peter H, Allison SD, Baho DL, Berga M, Bürgmann H, Huber DH, Langenheder S, Lennon JT, Martiny JBH, Matulich KL, Schmidt TM, Handelsman J. 2012. Fundamentals of microbial community resistance and resilience. *Front Microbiol* 3:417.
- Pimm SL. 1984. The complexity and stability of ecosystems. *Nature* 307:321–326.
- Orwin KH, Wardle DA. 2004. New indices for quantifying the resistance and resilience of soil biota to exogenous disturbances. *Soil Biol Biochem* 36:1907–1912.

We have now added this content as a new paragraph to the discussion

R2.3. Statistical reporting is inconsistent, and the figures should better represent the complex factorial design.

>>>Thank you. We have reviewed our statistical reporting throughout the Results and found it to be consistent. Briefly, we report the test applied, R/R2 and/or test statistic and P value. For post-hoc non-significant tests (e.g., Tukey's HSD) $p_{adj} > 0.05$ is reported. We also reviewed our figures and found that have reported complete elements of the experimental design across them (treatment, genotype, location, generation). If there is specific test, line, or figure that does not meet this expectation, please alert us and we are happy to update it.

R2.4. Explanation for the observed resistance, such as soil buffering, vermiculite use, has not been discussed.

>>>Thank you for this comment. We add a comment in the discussion about the possibility of unknown experimental aspects that may stabilize the microbiome.

R2.5. Abstract: Too broad and overstated. Rhizosphere microbiomes can be resistant to drought stress". Only one example of a legacy flavor effect is mentioned, creating a slight imbalance. Therefore, there is a need to reemphasize the genotype-specific and context-dependent nature of response, for example, "resistance to drought in the Red Hawk genotype across soils, while the Flavert genotype showed minor legacy effects

>>>Thank you for this comment. There was no context-dependency observed here, given that there was no or weak effects of drought or legacy or it. We remind that the Flavert did not show any legacy effects except in one instance (and the effects of this were very weak): the second generation in the rhizosphere which we explicitly mention in the abstract as the exception. We have added a clarifying phrase in the abstract to emphasize this.

R2.6. Introduction: I think there is a need to refine the hypothesis, to explicitly mention testing for both compositional shifts (beta diversity) and increased heterogeneity (beta dispersion) as indicators of disturbance, and to rewrite the introduction.

>>>Thank you for this comment. We have added these terms to the clarify this hypothesis in the Introduction, and also note that this terminology is already used in the Results and Methods sections.

7. Results: The problem of using different soils in G1 and G2 for the French experiment is a major confounding factor. Any "legacy" effect in France could be masked or altered by the completely new soil microbiome in G2. This severely limits the interpretation of the French G2 legacy results. Moreover, Different harvest timepoints (R5 in France vs. Senescence in Michigan) indicate the microbiomes were sampled at different ecological successional stages. A drought effect at R5 might recover by senescence, making direct comparisons of "effect size" between locations invalid. Different seed sterilization protocols indicate the initial microbial inoculum for plants was fundamentally different, indicating location effects beyond just the soil.

>>>Thank you for this note. We have discussed these differences between locations in the Methods. We do not attempt to directly compare effect sizes by locations, as there was no effect with only one very weak exception in the Flavert rhizosphere compartment in G2. This methodological information is available transparently for readers to assess the experiments independently in the Methods section.

8. Results. Often, interpret a non-significant PERMANOVA as "resistance." A non-significant result could be due to high variability, low statistical power, or a true lack of effect. The authors should say "we did not detect a significant effect of drought" rather than asserting "the microbiome was resistant." The beta-dispersion analysis helps here, but the language should be precise

>>> Thank you. This comment is unclear because: 1) we report a complete analysis of variance (beta dispersion as the reviewer asks), and 2) we applied a permutation based analysis on an appropriately-powered dataset for these analyses.

Furthermore, we have used "find and replace" to search for the words "resistance" or "resistant" within the results section and did not find any instances of their use therein. All statistics appear to be originally reported as "no effect of drought detected" or similar phrasing. Thus, we have already applied the precise language requested here.

9. I could not understand why some PERMANOVA R^2 values are very low (e.g., $R^2=0.03$ for Red Hawk G2 roots in France), even when significant ($P=0.029$). These should be highlighted as very weak effects, explaining almost none of the variance.

>>>We agree and this is exactly why we have consistently reported that there are either not detected or incredibly weak effects of drought.

10. Michigan G2 rhizosphere result is confusing: "a global interaction... was initially detected... but post-hoc pairwise tests did not support it." This suggests the overall model may be overfitted or underpowered, and the result should be treated as non-significant.

>>>We agree, and this is why we report and interpret it as non-significant.

11. Figures 5, 7, and 8 use color and shape to show a 2x2 factorial design (G1_G2), but it remains visually challenging to interpret. A different plot type (e.g., faceted bar plots for alpha diversity) might more clearly show the main and interaction effects.

>>> We did not detect significant or strong main or interaction effects, which is why the figures do not show clear patterns. We suggest that because there were not strong or statistically supported differences between categories, it is unlikely that a different visualization would show more pronounced differences. These visualizations were chosen because they show all aspects of the experimental design organized by the most relevant comparison.

12. Discussion: You have focused on the methodological differences as a strength for identifying generalities, but do not sufficiently acknowledge them as"

limitations for mechanistic interpretation." For example usage of a 50/50 soil-vermiculite mix, while justified for drainage, creates an artificial, low-carbon environment that likely reduces microbial biomass and buffers against stress, potentially causing the "resistance" observed. Again, overinterpretation or overstating of stability. For example, "microbiome resistance is an important component of ecosystem stability" is a broad ecological claim. The research shows a lack of response in one system; it does not provide evidence that this resistance is a general or adaptive feature that contributes to ecosystem function.

>>> Thank you for this comment.

>>> We did not detect notable differences in microbial community size across treatments, suggesting that there are limited concerns about biomass differences. The plants were also fertilized to ensure that nutrients were not limited.

>>>We respectfully disagree that we are overinterpreting or overstating stability. We have added the ecological theory around stability and provided additional references. The claim that “resistance is a component of stability” is not ours but is founded in decades of ecology and ecosystem sciences research, including the classic theoretical framework presented by Pimm in 1984. Our lab has been investigating microbiome disturbance responses and disturbance for nearly two decades and applying this ecological foundation.

>>>The research presented here shows an unchanged microbiome in more than one system: we performed experiments in two locations and with two genotypes and across two generations. While we agree and do not attempt to make a large scale generalization, we believe it is a fair interpretation that these results show overall resistance of the microbiome to drought in these experiments. We end with a simple suggestion that researchers consider both resistance and resilience in the future work. We have used cautious and tempered language throughout (e.g., the abstract “can be driven by stress resistance”).

13. I did not see the full report of the differential abundance analysis. While "fishing" is a concern, a standard, non-hypothesis-driven test for differentially abundant taxa (e.g., using DESeq2 or ANCOM-BC) is an expected part of microbiome papers.

>>>Please see our response to the previous reviews, which includes a detailed response about why we do not feel it is appropriate to overextend the data in this way due to the non-significant effects of drought.

14. I could not find the reasons for the critical method differences between locations (especially the different G2 soil in France and the different harvest

timepoints). This sentence is good: "A different soil was used in G2 to simulate..." is good, but why wasn't this done for Michigan?

>>>The researchers in Michigan had access to the same soil plots for both generations, while the French researchers did not.

Minor comments

R2.15

1. The research is not simply about microbiome resistance; it is actually about "Microbiome resistance to drought is genotype-specific and context-dependent, and multi-generational legacy effects are minimal or absent in the systems we tested. The manuscript should reflect this more nuanced conclusion.

>>>We respectfully disagree with this statement, which is contrast to the reviewer's previous comment: As the reviewer pointed out (R2.9), the effects of drought were non-significant or very weak, with post-hoc tests often showing no consistent effects. Thus, there is not much nuance to report here. We openly and completely report the lack of drought effects and lack of legacy effects without overextending the data.

R2.16

2. Use consistent language throughout the manuscript ("no significant effect was detected") and evidence of absence ("the community was resistant"). Report effect sizes (R^2) alongside p-values will not help to judge the biological importance of statistical results.

>>>As per response to R2.8 and R2.3.

Suggestions

R2.17

1. To verify claims of functional stability, perform any of the resistance genes only on archived samples, to test whether the microbiome shows transcriptional resistance or activates another stress response pathway.

>>>This comment is unclear. We did not assess resistance genes in this work. We did not collect RNA from this study and cannot perform transcriptional analysis.

R2.18

2. It might be that the resistance is an emergent property of native soil or an artificial soil plus vermiculite mixture. Please see this

>>>This comment is unclear. Resistance cannot be a property of vermiculite. It is a component of stability that is calculated from systems, like population, communities or ecosystems.

R2.18

3. If you have time, I suggest introducing a 3rd-generation challenge with a more severe drought to unmask any potential cryptic legacy effects on plant resilience that were not evident from compositional microbiome data alone. Please think about this

>>>Thank you, but to add a third generation of this factorial experiment is not a reasonable request (this would involve hundreds of plants, over a year of additional work, time of two technicians or students, and a sequencing budget that exceeds what we have already had for G0-G2).

>>>We find this request to be confusing especially given that there were no notable experimental effects in microbiome in G0-G2 and assess there is a high risk of no new/additional information being generated with such an immense effort.

(What are “cryptic legacy effects”?)

Re: Spectrum03019-25R2 (**Evaluating the legacy of drought exposure on root and rhizosphere bacterial microbiomes over two plant generations**)

Dear Prof. Ashley Shade:

Your manuscript has been accepted, and I am forwarding it to the ASM production staff for publication. Your paper will first be checked to make sure all elements meet the technical requirements. ASM staff will contact you if anything needs to be revised before copyediting and production can begin. Otherwise, you will be notified when your proofs are ready to be viewed.

Sincerely,
Weimin Sun
Editor
Microbiology Spectrum